# Light-sensing via hydrogen peroxide and a peroxiredoxin

Kristofer Bodvard[1,2,*], Ken Peeters[1,*], Friederike Roger[1,*], Natalie Romanov[3,†], Aeid Igbaria[4], Niek Welkenhuysen[1,5], Gaël Palais[4], Wolfgang Reiter[6], Michel B. Toledano[4], Mikael Käll[2] & Mikael Molin[1]

Yeast lacks dedicated photoreceptors; however, blue light still causes pronounced oscillations of the transcription factor Msn2 into and out of the nucleus. Here we show that this poorly understood phenomenon is initiated by a peroxisomal oxidase, which converts light into a hydrogen peroxide ($H_2O_2$) signal that is sensed by the peroxiredoxin Tsa1 and transduced to thioredoxin, to counteract PKA-dependent Msn2 phosphorylation. Upon $H_2O_2$, the nuclear retention of PKA catalytic subunits, which contributes to delayed Msn2 nuclear concentration, is antagonized in a Tsa1-dependent manner. Conversely, peroxiredoxin hyperoxidation interrupts the $H_2O_2$ signal and drives Msn2 oscillations by superimposing on PKA feedback regulation. Our data identify a mechanism by which light could be sensed in all cells lacking dedicated photoreceptors. In particular, the use of $H_2O_2$ as a second messenger in signalling is common to Msn2 oscillations and to light-induced entrainment of circadian rhythms and suggests conserved roles for peroxiredoxins in endogenous rhythms.

[1] Department of Chemistry and Molecular Biology, University of Gothenburg, Box 462, S-413 90 Göteborg, Sweden. [2] Department of Physics, Chalmers University of Technology, S-412 96 Göteborg, Sweden. [3] Mass Spectrometry Facility, Max F. Perutz Laboratories, University of Vienna, Dr Bohrgasse 9, A-1030 Vienna, Austria. [4] Oxidative Stress and Cancer, SBIGEM, iBiTec-S, FRE3377 CEA-CNRS-Université Paris-Sud, CEA-Saclay, bat 142 F-91191 Gif Sur Yvette, France. [5] Hohmann Lab, Department of Biology and Biological Engineering, Chalmers University of Technology, S-412 96 Göteborg, Sweden. [6] Department of Biochemistry, Max F. Perutz Laboratories, University of Vienna, Dr Bohrgasse 9, A-1030 Vienna, Austria. * These authors contributed equally to this work. † Present address: Department of Structural and Computational Biology, EMBL, Meyerhofstrasse 1, 691 17 Heidelberg, Germany. Correspondence and requests for materials should be addressed to M.M. (email: mikael.molin@cmb.gu.se).

Sunlight is a prerequisite for life by providing a source of heat and energy for primary production. Most organisms exposed to light therefore maintain an ability to adapt their activities to its presence. For example, many organisms, from cyanobacteria to humans[1], maintain autonomous cycles of activity that correspond to day and night even in complete darkness[2]. The ability to respond to light is a fundamental feature of such circadian clocks[3]. In the yeast Saccharomyces cerevisiae, light alters ultradian metabolic rhythms[4], which have been proposed to constitute models for circadian clocks[5]. In addition, the Zn-finger transcription factor Msn2 rhythmically shuttles into and out of the nucleus in response to illumination[6], in spite of the lack of specialized photosensory proteins such as opsins, phytochromes and cryptochromes in this organism[7]. Nuclear localization of Msn2 is inhibited by cyclic AMP-controlled protein kinase A (PKA)[8,9] and stimulated by the phosphatases PP1 and PP2A[10,11]. The Msn2 rhythm was proposed to originate from oscillations in cAMP produced at intermediate light intensities[12], based on the observation that Msn2 fails to oscillate in mutants deficient in PKA feedback regulation[13] and on the recapitulation of oscillations in a mathematical model of the PKA signalling pathway. A careful analysis of the localization behaviour in single cells indicated that Msn2 localization typically passes through three distinct and successive states in response to illumination: cytoplasmic, nucleo-cytoplasmic oscillatory and more permanently nuclear[14]. Such gradually increased nuclear localization of Msn2 presumably reflects a gradual decrease in PKA-dependent phosphorylation and altered levels of PKA/phosphatase signalling intermediate(s). However, the nature of any such intermediates and the mechanism by which PKA senses light remains unknown.

Typical 2-Cys peroxiredoxins are enzymes that reduce hydrogen peroxide ($H_2O_2$) via two catalytic cysteines and thioredoxin acting as the hydrogen donor[15,16]. During catalysis, a small proportion of the primary catalytic (peroxidatic) cysteine becomes hyperoxidized to the sulfinylated (Cys–$SO_2H$) form, which inactivates the enzyme[17,18]. Peroxiredoxins have received increased attention because of their functions in suppressing malignant tumours[19], in keeping the genome free of mutations[20] and as conserved executors of the ability of caloric restriction to slow down the rate of ageing[15,16,21–24]. In addition, peroxiredoxin hyperoxidation was shown to sustain transcription-independent circadian rhythms in organisms from the three kingdoms of life[1,25–27]. On a similar note, oscillations in peroxiredoxin hyperoxidation were shown to resonate with metabolic tidal rhythms in the marine crustacean Eurydice pulchra and peroxiredoxins have also recently been suggested to modulate yeast ultradian metabolic rhythms[5,28], indicating conserved functions in maintaining endogenous molecular clocks. The observation that light stimulates $H_2O_2$ production in cultured mouse, monkey and humans cells via photoreduction of flavin-containing oxidases such as peroxisomal acyl-coenzyme A (CoA) oxidase[29] and the messenger function of $H_2O_2$ in zebrafish, which couples the light signal to the circadian clock[30], raise the interesting possibility that $H_2O_2$ might be a conserved second messenger by which light affects cellular physiology in general and endogenous clocks in particular.

Here we show that in S. cerevisiae a conserved peroxisomal oxidase converts the light impulse into a $H_2O_2$ signal that is sensed by the peroxiredoxin Tsa1 and then transduced to thioredoxin to inhibit PKA activity. We propose that peroxiredoxin-mediated $H_2O_2$ signalling establishes rhythmic Msn2 nuclear accumulation by superimposing on PKA feedback regulation. In particular, we report that the Tsa1-mediated signal counteracts the nuclear retention of the two most highly expressed of the PKA catalytic subunits, thereby antagonizing a process that contributes to delayed Msn2 nuclear localization in response to $H_2O_2$.

## Results

### Light is signalled to Msn2 through $H_2O_2$.
The transcription factor Msn2 rhythmically shuttles into and out of the nucleus in response to light[6]. To study this phenomenon, we used an experimental setup enabling both single-cell (Fig. 1a–c and Supplementary Movie 1) and population analyses (Fig. 1d) of Msn2 localization dynamics[14]. In short, cells were exposed to blue visible light ($\lambda = 450–490$ nm) and imaged in a perfusion chamber using epifluorescence microscopy as previously described[14]. At a light intensity corresponding to that on a sunny day ($P = 115 \mu W$), Msn2 accumulated into the nucleus and then back to the cytoplasm on average $4.3 \pm 3.6$ times per hour (average ± s.d., $n = 248$) and, following an initial lag of $\sim 20$ min, the proportion of cells exhibiting Msn2 nuclear localization[14] stabilized at around 25% (Fig. 1d). Higher light intensities ($P = 200 \mu W$) lead to a higher proportion of cells exhibiting nuclear localization (Supplementary Fig. 1a) and eventually resulted in more sustained nuclear localization[14,31]. In the absence of light, Msn2 did not accumulate into the nucleus[6].

We hypothesized that $H_2O_2$ might serve as the light-induced second messenger, because light stimulates $H_2O_2$ production in mammalian cells[29] and because yeast cells lacking the oxidant-responsive transcription factor Yap1 grow poorly in light[4]. To substantiate this, we used the cytosolic genetically encoded $H_2O_2$ sensor HyPerRed[32], which we reasoned would be less sensitive to bleaching upon illumination with blue light than green fluorescent protein (GFP)-based $H_2O_2$ probes (for example, HyPer[33] and roGFP2-Tsa2$\Delta CR$[34]). Upon the start of illumination, the HyPerRed signal initially stayed constant but already after a couple of minutes decreased steadily at a rate dependent on light intensity (Supplementary Fig. 1b,c), indicating that probe bleaching interfered with its use in $H_2O_2$ determination. Nevertheless, in cells expressing an $H_2O_2$-insensitive probe (carrying a C199S substitution in the OxyR $H_2O_2$-sensing domain[32]), fluorescence decreased faster and/or to lower levels than in cells expressing the $H_2O_2$-sensitive probe (Supplementary Fig. 1b,c), indicating that upon illumination the levels of cytosolic $H_2O_2$ increased (Fig. 1e). The Cys199-dependent HyPerRed fluorescence increased as a function of light intensity (Fig. 1e), suggesting that so do the levels of cytosolic $H_2O_2$. In support of $H_2O_2$ causing Msn2 nuclear redistribution upon illumination, an increased number of cells lacking the mitochondrial cytochrome c peroxidase Ccp1 exhibited nuclear Msn2 (Fig. 1d and Supplementary Fig. 1d,e); these cells are deficient in $H_2O_2$ scavenging. In addition, in wild-type cells, Msn2 rhythmically shuttled between the nucleus and the cytoplasm upon addition of intermediate levels of $H_2O_2$ (0.4 mM, Fig. 1f) and this level of exogenous $H_2O_2$ produced a comparable, albeit more rapid, increase in cytosolic $H_2O_2$ (C199-dependent HyPerRed fluorescence, Supplementary Fig. 1f,g). Higher levels of $H_2O_2$ elicited sustained localization of Msn2 to the nucleus (Supplementary Fig. 1h–j). Conversely, in cells overproducing Ccp1 ($\sim 3$-fold, Supplementary Fig. 1k,l), which scavenge $H_2O_2$ at an increased rate, light-induced Msn2 nuclear localization was abolished (Fig. 1g), suggesting that cytosolic $H_2O_2$ is both necessary and sufficient for the light response of Msn2.

### A peroxisomal oxidase converts light into a $H_2O_2$ signal.
Flavin prosthetic groups play unique roles in redox reactions by their ability to participate in both one- and two-electron transfer processes[35]. In addition, they act as chromophores in dedicated

light receptors of the 'light-oxygen-voltage', the 'blue-light sensing using Flavin' and the cryptochrome families[36]. In these proteins, photoexcitation of the flavin group initiates signal transduction through intramolecular conformational changes and/or altered protein–protein interactions[36]. However, upon excitation, flavins also become sensitive to reduction by cellular reducing agents and can subsequently transfer electrons to molecular oxygen, leading to the production of $H_2O_2$ (refs 35,37,38). Such reactions of flavin in peroxisomal fatty-acyl CoA oxidase were proposed to be the cause of blue light phototoxicity in mammalian cells through $H_2O_2$ production[29]. The yeast homologue of peroxisomal acyl-CoA oxidase is Pox1. Strikingly, Pox1 deficiency substantially decreased both cytosolic $H_2O_2$ (Fig. 1h) and Msn2 nuclear localization upon illumination (Fig. 1d and Supplementary Fig. 1m,n). However, both cytosolic $H_2O_2$ and Msn2 nuclear localization increased normally upon the addition of exogenous $H_2O_2$ to cells lacking Pox1 (Supplementary

Fig. 1o–r), indicating that peroxisomal acyl-CoA oxidase affects the response to light via $H_2O_2$. These data suggest that a peroxisomal acyl-CoA oxidase can signal light through $H_2O_2$ production in an organism lacking dedicated light receptors.

**A peroxiredoxin acts as a receptor of light-induced $H_2O_2$.** The peroxiredoxin Tsa1 (see Fig. 2a) is required for the nuclear accumulation of Msn2 in response to $H_2O_2$, but not to other stimuli, for example, NaCl that also trigger this response[39]. Based on our finding that light uses $H_2O_2$ as a messenger, we therefore asked whether light-induced Msn2 nuclear localization similarly depended on Tsa1. Indeed, we found that Msn2 remained cytoplasmic upon illuminating Tsa1-deficient cells, as evidenced by a representative single-cell trace (Fig. 2b) and by the strongly decreased fraction of cells displaying Msn2 nuclear localization (Fig. 2c). This was true also at higher light intensities ($P = 200 \mu W$, Supplementary Fig. 2a), further underscoring the crucial role of $H_2O_2$ in yeast light sensing. Moreover, we found that Tsa1 catalytic cycling was required for signal transduction, as Msn2 remained cytosolic in cells carrying serine substitutions of the Tsa1 catalytic cysteines (Fig. 2a,c,d and Supplementary Fig. 2b–d). It is known that a small fraction of Tsa1 becomes hyperoxidized to the sulfinic acid form upon each turn of the peroxiredoxin catalytic cycle and such hyperoxidation inactivates enzyme peroxidatic cycling[17,18] (see Fig. 2a). To further test the importance of enzyme cycling, we therefore analysed Msn2 localization in cells expressing *tsa1ΔYF*. This mutant lacks the carboxy-terminal YF motif and is therefore resistant to hyperoxidation[40,41] (Supplementary Fig. 2k). The fraction of *tsa1ΔYF* cells displaying Msn2 nuclear accumulation increased markedly (Fig. 2d and Supplementary Fig. 2e). However, this effect was completely negated by serine substitution of C171, which decreased Msn2 nuclear accumulation back to the low levels observed in the *tsa1C171S* single mutant (Fig. 2d and Supplementary Fig. 2f). These observations thus further point to the importance of Tsa1 peroxidatic cycling for light signalling to Msn2 and the inhibitory effect of enzyme hyperoxidation on this response.

Tsa1 might relay the $H_2O_2$ signal produced by light, by oxidizing an Msn2 regulatory target either directly or through the intermediacy of its dedicated reductase, the redundant cytosolic thioredoxins Trx1 and Trx2. As the latter are required for $H_2O_2$ signalling to Msn2 (ref. 39), we tested whether they were required also in light signalling. We found that simultaneous inactivation

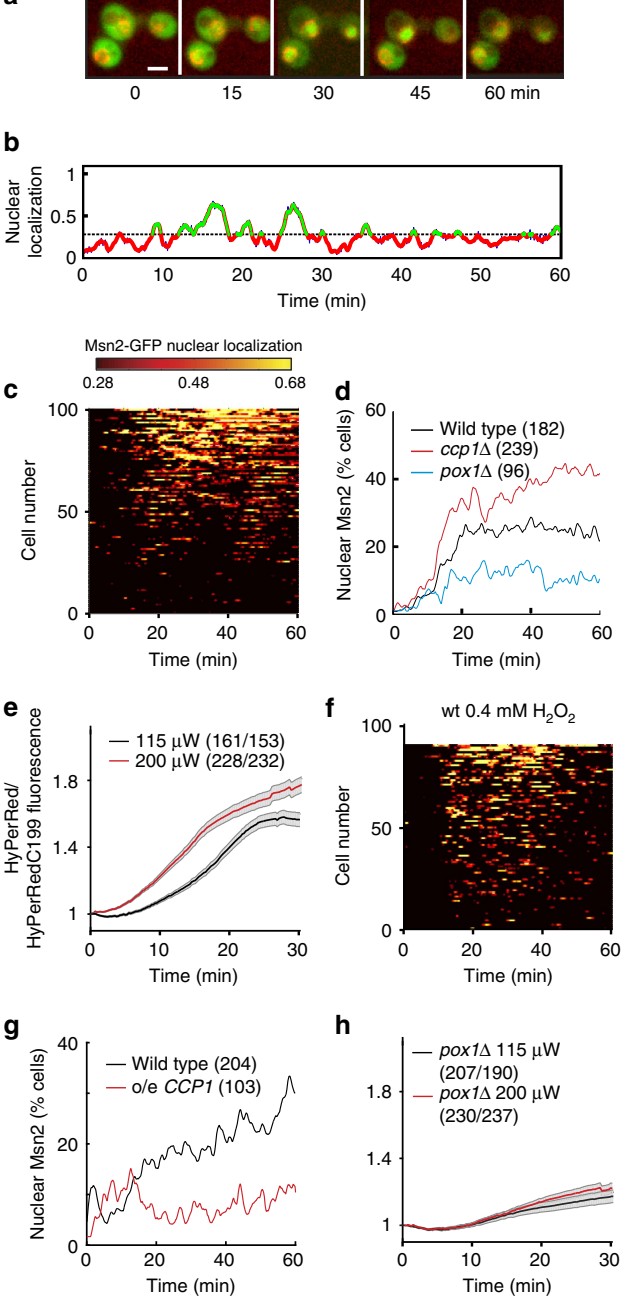

**Figure 1 | $H_2O_2$ is necessary and sufficient for light-induced nuclear translocation of Msn2.** (**a**) Frames from Supplementary Movie 1 showing that Msn2-GFP (green) rhythmically shuttles into and out of the nucleus upon illumination of cells with blue light (intensity 115 μW). Nup49-mCherry (red) was used to visualize the nucleus. Scale bar, 5 μm. (**b**) Single-cell trace representing nuclear localization upon illumination of a wild-type (wt) cell, quantified by comparing the signal intensity in the nucleus with the signal intensity in the cytosol (see Methods). The dashed line indicates the threshold ratio of nuclear/cytosolic Msn2 signal for nuclear localization in population analyses[14]. (**c,f**) Msn2 nucleocytoplasmic localization trajectories in individual wt cells upon illumination (115 μW, **c**) or upon the addition of 0.4 mM $H_2O_2$ (**f**). The colour scale indicates the degree of nuclear localization. (**d,g**) Fraction of wild-type, *ccp1Δ* and *pox1Δ* cells (**d**) or cells overexpressing (o/e) *CCP1* or not (**g**) displaying nuclear localization of Msn2 following illumination at 115 μW. The number of cells assayed in each analysis are indicated in brackets. (**e,h**) Cytosolic $H_2O_2$, measured as average Cys199-dependent HyPerRed fluorescence, upon illumination of wt (**e**) and *pox1Δ* cells (**h**) at the indicated light intensities. Error bars indicate s.e.m. The number of HyPerRed/ HyPerRedC199S cells assayed in each analysis are indicated in brackets.

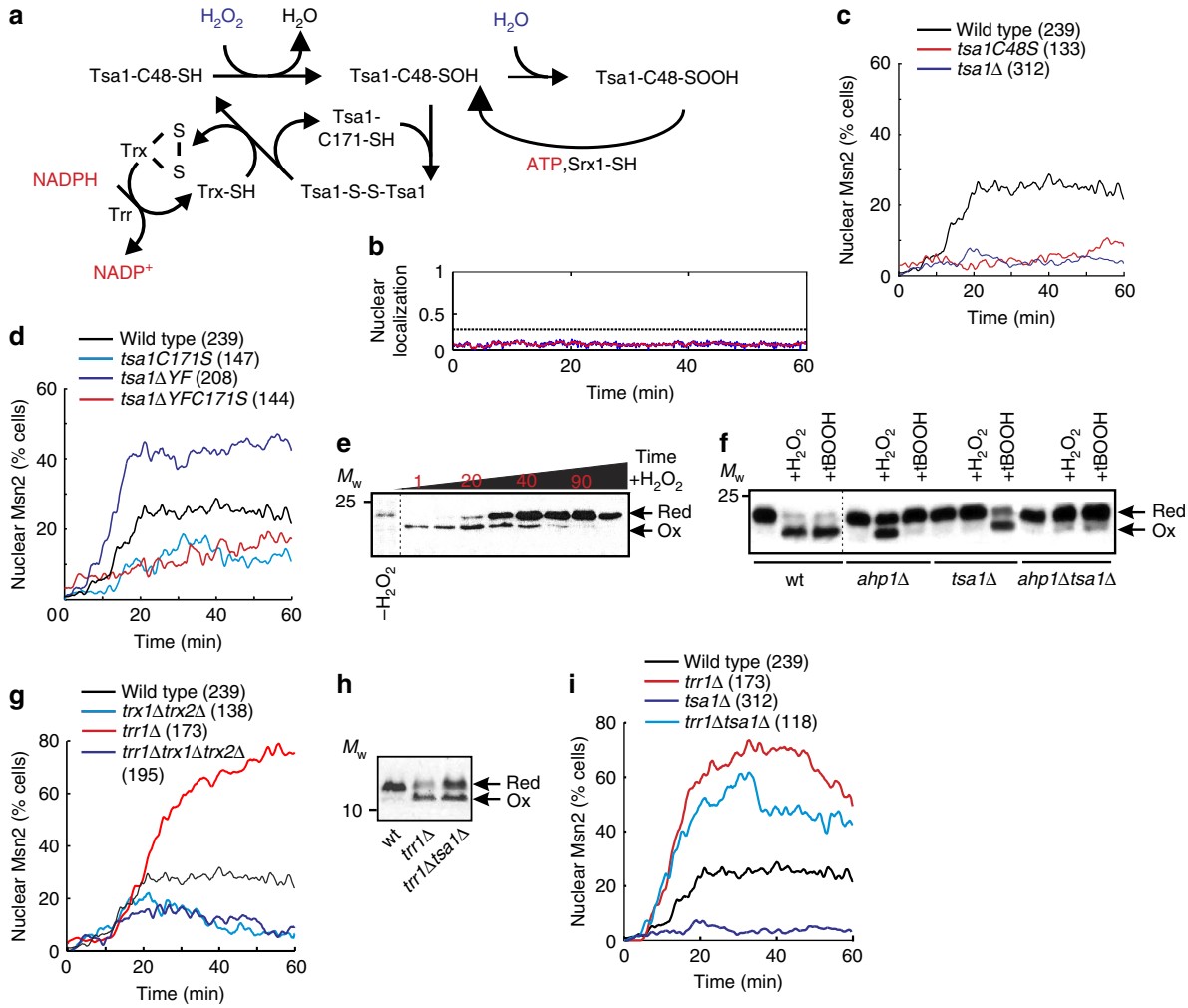

**Figure 2 | The light-induced H$_2$O$_2$ signal is transduced by the peroxiredoxin Tsa1 which instantly relays the signal to the cytosolic thioredoxins.**
(**a**) Overview of catalysis in Tsa1, a typical two-cysteine peroxiredoxin. H$_2$O$_2$ is reduced by Tsa1-C48-SH, which is oxidized to a sulfenic acid intermediate (Tsa1-C48-SOH), which subsequently condenses with the resolving cysteine (C171-SH) of another Tsa1 molecule, producing a disulfide-bonded Tsa1 dimer. This disulfide bond is subsequently reduced by thioredoxin (Trx), completing the catalytic cycle. A proportion of C48-SOH becomes hyperoxidized to the sulfinic acid form (C48-SOOH) in each catalytic cycle, inactivating peroxidase activity. Sulfiredoxin (Srx1) reduces C48-SOOH, allowing Tsa1 to re-enter the catalytic cycle. (**b**) Single-cell trace of Msn2 nuclear localization in a typical *tsa1Δ* cell following illumination at 115 µW. (**c,d,g,i**) Fraction of cells of the indicated strains displaying nuclear localization following illumination at 115 µW. The number of cells assayed in each analysis are indicated in brackets. (**e**) Trx2 oxidation assayed by redox immunoblot analysis of a strain expressing ProteinA-tagged Trx2. Samples for redox immunoblot analysis were withdrawn before and 1, 10, 20, 30, 40, 60, 90 and 120 min following H$_2$O$_2$ addition (0.3 mM). Red, reduced form; Ox, oxidized form. See also Supplementary Fig. 2l. (**f**) Trx2 oxidation in Tsa1- and Ahp1-deficient cells upon H$_2$O$_2$ addition (0.3 mM for 1 min) or the addition of *t*-butylhydroperoxide (*t*BOOH, 0.3 mM for 1 min). (**h**) Thioredoxin oxidation in cells lacking Trr1 or both Trr1 and Tsa1. See also Supplementary Fig. 2o.

of Trx1 and Trx2 (*trx1Δtrx2Δ*) impaired light signalling to Msn2 (Fig. 2g and Supplementary Fig. 2b,g). Upon reducing peroxiredoxins, thioredoxins become oxidized and are then reduced by thioredoxin reductase (see Fig. 2a). Trx2 was indeed immediately oxidized in a Tsa1-dependent manner upon H$_2$O$_2$ addition (Fig. 2e,f and Supplementary Fig. 2l,m). The other abundant peroxiredoxin Ahp1 is also recycled by the cytosolic thioredoxins but Ahp1 preferentially reduces the model organic peroxide *t*-butyl hydroperoxide and therefore caused Trx2 oxidation in response to this particular oxidant but not to H$_2$O$_2$ (Fig. 2f). In contrast, Tsa1 preferentially reduces H$_2$O$_2$ and caused Trx2 oxidation in response to H$_2$O$_2$ but not to *t*-butylhydroperoxide (Fig. 2f), suggesting a high degree of specificity in H$_2$O$_2$-Tsa1-thioredoxin-mediated signal transduction. To probe whether the thioredoxins are required for signalling when in their oxidized form, we used a mutant lacking the cytosolic thioredoxin reductase (*TRR1*) in which thioredoxins are

about 70% oxidized already before the addition of H$_2$O$_2$ (ref. 42) (Fig. 2h and Supplementary Fig. 2n,o). The *TRR1* mutation strongly increased Msn2 nuclear localization in response to light (Fig. 2g and Supplementary Fig. 2h) and H$_2$O$_2$ (ref. 39). However, this increased response was completely lost upon removing both cytosolic thioredoxins (*trr1Δtrx1Δtrx2Δ*, Fig. 2g and Supplementary Fig. 2i). In this mutant, Msn2 nuclear localization was brought back to the low levels observed in the *trx1Δtrx2Δ* mutant, establishing that the thioredoxins signal light in their oxidized form. High levels of oxidized Trx2 were maintained in cells lacking both Trr1 and Tsa1 (*trr1Δtsa1Δ*, Fig. 2h and Supplementary Fig. 2o), enabling us to investigate whether the lack of response in Tsa1-deficient cells was caused by deficient Trx2 oxidation. Indeed, in *trr1Δtsa1Δ* cells, light triggered a vigorous Msn2 response (Fig. 2i and Supplementary Fig. 2j), suggesting that Tsa1 becomes dispensable for light signalling upon the accumulation of thioredoxins in their oxidized form.

Taken together, these results suggest that Tsa1 functions as a specific receptor for $H_2O_2$, but not organic peroxides, and transduces a signal to the cytosolic thioredoxins. This oxidation cue is both necessary and sufficient for Msn2 nuclear localization in response to illumination.

**Light inhibits Msn2 phosphorylation by PKA via a $H_2O_2$ relay.** Light-induced Msn2 nuclear accumulation was prevented in cells with constitutive PKA activity ($bcy1\Delta$ or $pde2\Delta$, Fig. 3a), as shown previously[12,14], and, importantly, the same was true for $H_2O_2$ at levels that recapitulated the light response (0.4 mM, Supplementary Fig. 3a,b). However, the Msn2 response to high levels of $H_2O_2$ (0.8 mM) was not inhibited by increased PKA activity (Supplementary Fig. 3b). On the contrary and in agreement with previously published data[39], the response to high levels of $H_2O_2$ was stimulated in cells expressing constitutive PKA activity, indicating that (an)other pathway(s) operates under this condition. At these levels of $H_2O_2$, Tsa1 was still required for the Msn2 response[39] (Supplementary Fig. 3c), whereas upon further increased $H_2O_2$ (1 mM), it was not (Supplementary Fig. 3c), suggesting that the requirement for Tsa1 could be overcome upon increased $H_2O_2$.

PKA excludes Msn2 from the nucleus by both inhibiting its nuclear import and by stimulating its nuclear export through phosphorylation of serine residues in the nuclear localization (NLS) and nuclear export (NES) sequences[9,43,44]. We found that a fusion construct containing a heterologous NES and the Msn2 NLS (amino acids 576–704), which becomes dephosphorylated and concentrates into the nucleus upon glucose starvation[43],

neither responded to light nor $H_2O_2$ (0.4 mM, Supplementary Fig. 3d), suggesting that these cues regulate Msn2 at the level of nuclear export. In agreement with this, alanine substitution of one or both of the PKA-cognate NES residues S288 and S304 enhanced light-induced nuclear localization of Msn2 in wild-type cells and was able to restore nuclear localization in cells with constitutive PKA activity ($bcy1\Delta$ or $pde2\Delta$, Fig. 3a–c and Supplementary Fig. 3e). These NES mutations also restored the Msn2 light (Fig. 3a–c and Supplementary Fig. 3f,g) and $H_2O_2$ responses (Supplementary Fig. 3h,i) in cells lacking Tsa1, suggesting that light inhibits Msn2 nuclear export by counteracting the phosphorylation of these serine residues. The Tsa1-dependent regulatory function on these two serine residues in response to light and $H_2O_2$ was further demonstrated by their rapid, dose-dependent and transient dephosphorylation upon exposure to $H_2O_2$ in wild-type cells, but not in cells lacking *TSA1*, as evidenced by both quantitative phosphoproteomic mass spectrometry (MS) analyses of SILAC (stable isotope labelling with amino acids in cell culture)-labelled purified Msn2 (Fig. 3d–f, Supplementary Fig. 3j–m and Supplementary Data 1 and 2) and immunoblot analyses of immunoprecipitated Msn2-NES using an antibody recognizing phosphorylated PKA target sequences (Fig. 3g,h and Supplementary Fig. 3n). Additional PKA-cognate Msn2 serine residues (S194, S451, S582 and S633 (ref. 44)) were also dephosphorylated in a Tsa1-dependent manner upon exposure to $H_2O_2$ (Supplementary Fig. 3j and Supplementary Data 1 and 2). Dephosphorylation of serine residues located in the Msn2 NLS (S582, S620, S625 and S633) might contribute to Msn2 nuclear localization upon illumination at higher light intensities and more severe $H_2O_2$

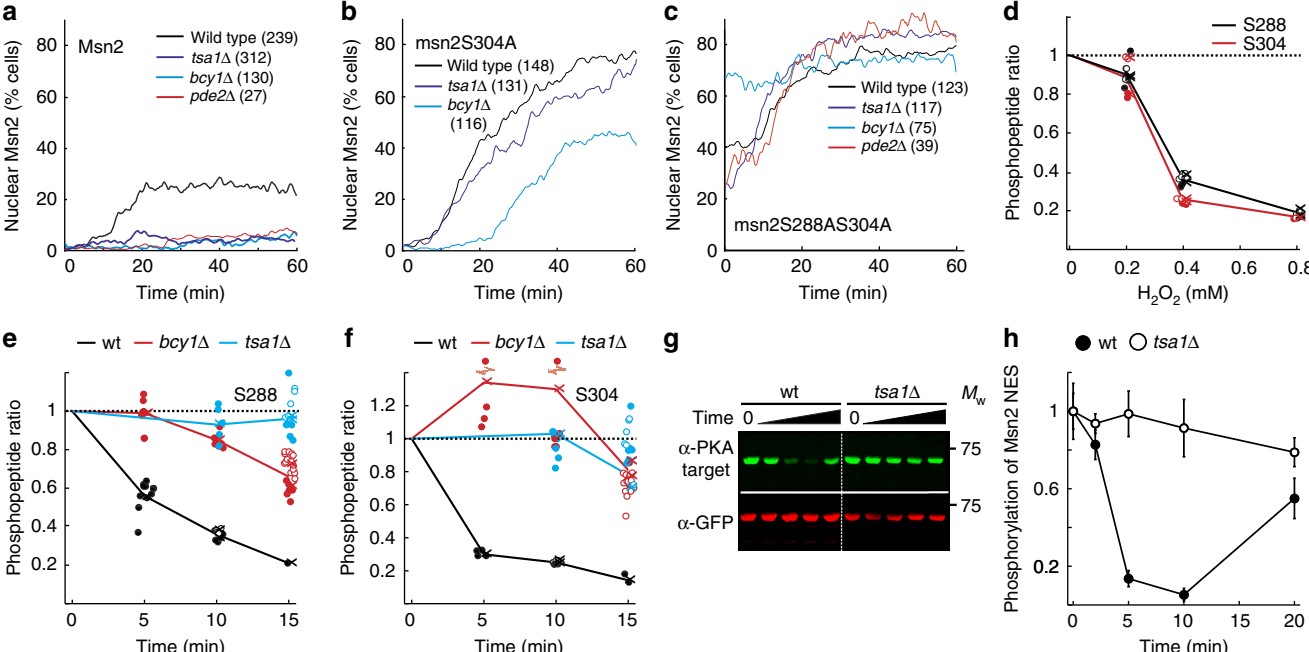

**Figure 3 | Tsa1 controls Msn2 translocation to the nucleus upon illumination and $H_2O_2$ by reducing Msn2 phosphorylation.** (**a–c**) Fraction of cells of the indicated strains displaying nuclear localization following illumination at 115 µW. Cells were transformed with plasmids carrying either wild-type or mutant Msn2 alleles as indicated. The number of cells assayed in each analysis are indicated in brackets. (**d–f**) SILAC-based MS quantification of Msn2 phosphopeptides spanning the indicated NES serines following the addition of the indicated amount of $H_2O_2$ for 10 min to the wild type (wt) (**d**) or 0.4 mM $H_2O_2$ for the indicated time to the indicated strains (**e,f**). Average ratios ($H_2O_2$-treated/untreated) of Msn2 phosphopeptides measured by quantitative MS in two biological replicates are shown, the individual values of which each is indicated by a cross. Closed and open circles denote individual peptide ratios quantified in biological replicate 1 and 2, respectively (see Methods and Supplementary Data 1 and 2 for details). (**g,h**) Msn2 NES phosphorylation determined by immunoblot analysis of immunoprecipitated Msn2 expressed from plasmid pAMG9 (ref. 8) and an antibody recognizing phosphorylated PKA target sequences. The average phosphorylation signal on the membranes (**g**) was quantified by densitometric analysis and normalized to the anti-GFP signal (**h**, $n = 3$). Error bars indicate s.d. See also Supplementary Fig. 3n.

treatment, as the *msn2S288AS304A* mutant still responded to illumination (Fig. 3c), whereas a mutant in which all six relevant NES and NLS serine residues[9,43] were substituted for alanine residues displayed permanent, light-independent nuclear accumulation (Supplementary Fig. 3o). In support of the role of Tsa1-dependent light signalling in counteracting PKA-dependent inhibition of Msn2, overexpression of the cAMP phosphodiesterase *PDE2*, which decreases PKA activity[45], restored the defective light response of the *tsa1Δ* mutant (Fig. 4a and Supplementary Fig. 4a–d), but did not restore the oxidation of thioredoxins (Supplementary Fig. 4g,h). In addition, thioredoxin oxidation was unaffected upon the addition of $H_2O_2$ in Tsa1-proficient cells overexpressing *PDE2*, further supporting Msn2 signal control downstream of the thioredoxins upon Pde2 overproduction (Supplementary Fig. 4e,f). Conversely, the increased light response in cells carrying the *tsa1ΔYF* mutant could be mitigated by overexpressing a PKA catalytic subunit Tpk1 (Fig. 4b and Supplementary Fig. 4i–m). Taken together, our data establish that the light-induced $H_2O_2$ signal transduced by Tsa1 primarily relieves Msn2 from inhibitory phosphorylation of serine residues in the nuclear export sequence, allowing Msn2 to accumulate in the nucleus.

**Intermediate PKA activity recapitulates Msn2 oscillations.** The use of a cell-permeable substrate analogue inhibitor (1-NM-PP1) in cells carrying substrate analogue-sensitive PKA catalytic

subunits (*tpk1M164Gtpk2M147Gtpk3M165G*[46]) provided a tool to further dissect the importance of PKA inhibition in light-induced Msn2 nuclear localization. Partial inhibition of PKA recapitulated intermediate and oscillatory Msn2 nuclear localization (Fig. 4c,d and Supplementary Fig. 4n), whereas a more drastic inhibition led to instant and permanent Msn2 nuclear localization (Fig. 4c and Supplementary Fig. 4o,p). The Msn2 response to partial PKA inhibition was, however, significantly faster than the response to both illumination and $H_2O_2$ (compare the response times to a level of inhibitor that recapitulated intermediate and oscillatory Msn2 nuclear localization (100 nM), 115 μW light or 0.4 mM $H_2O_2$, Fig. 4e), suggesting that the response delay reflects the time needed for $H_2O_2$ signalling to PKA.

**Tsa1 governs Msn2 through PKA cellular localization.** So far, our data indicate that light signalling prevents PKA-dependent inhibition of Msn2 nuclear export by counteracting the phosphorylation of NES residues. To further decipher how the light pathway modulates PKA activity, we monitored PKA subcellular location, using GFP fusions of its catalytic subunits. In agreement with published data[47,48], PKA catalytic subunits concentrated in the nucleus in the absence of $H_2O_2$ (Fig. 4f–h and Supplementary Fig. 4q–s), in a manner dependent upon the PKA regulatory subunit Bcy1 (Fig. 4f–h and Supplementary Fig. 4r). In contrast, upon $H_2O_2$ treatment the two most highly expressed

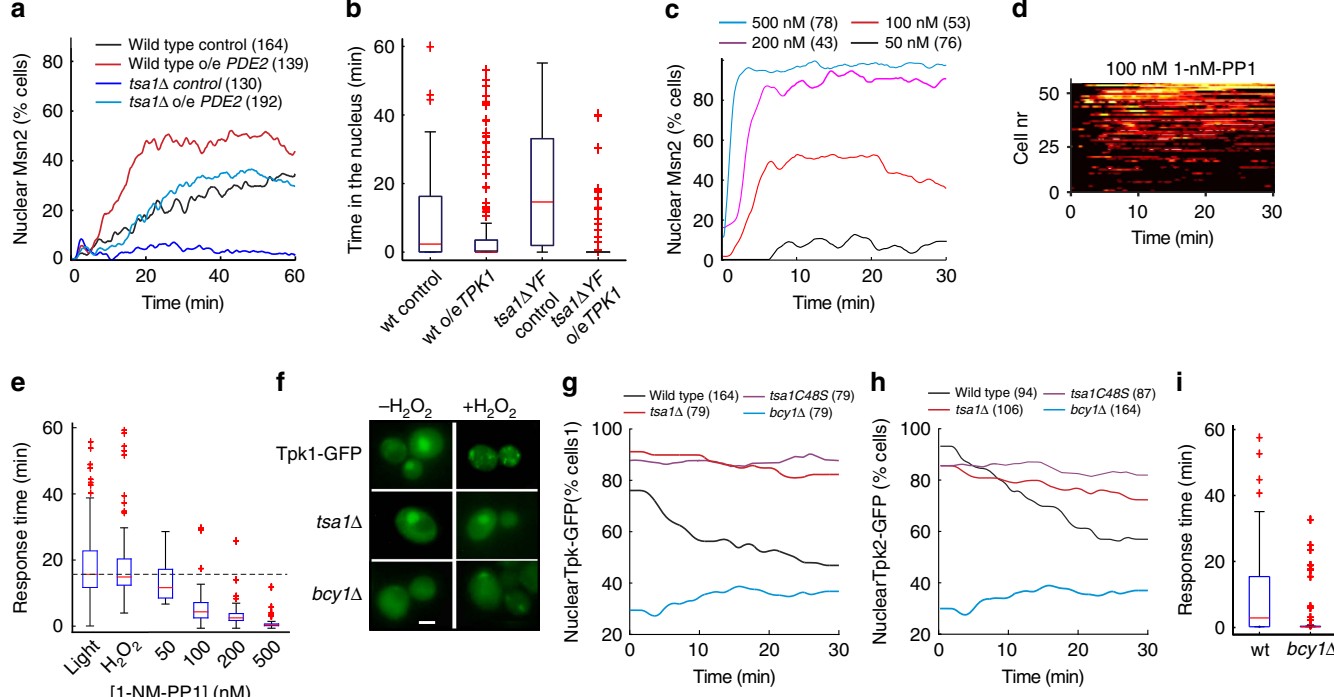

**Figure 4 | Tsa1 partially inhibits PKA and mediates Msn2 response delay through slow nuclear export of PKA catalytic subunits upon $H_2O_2$.**
(**a**) Fraction of wt or *tsa1Δ* vector control cells or cells overexpressing (o/e) *PDE2* (to induce low PKA activity) displaying nuclear localization following illumination at 115 μW. Red lines denote median values, boxes 25th and 75th percentiles and whiskers minimum and maximum values of data points not classified as outliers in MATLAB. (**c**–**d**) The number of cells assayed in each analysis are indicated in brackets. (**b**) Time Msn2 stayed in the nucleus following illumination of the wt or the *tsa1ΔYF* mutant o/e or not o/e *TPK1* (high PKA activity) at a light intensity of 115 μW. Fraction of cells displaying nuclear localization (**c**) nucleocytoplasmic localization trajectories (**d**) or time to first nuclear localization (**e**) of Msn2 in a strain carrying analog-sensitive PKA catalytic subunits (*tpk1M164G tpk2M147G tpk3M165G*[46]) treated with the indicated concentrations of an inhibitory substrate analog (1-NM-PP1).
(**f**) Localization of a Tpk1-GFP fusion protein in the indicated mutant strains in the absence of $H_2O_2$ ( − $H_2O_2$) or 6 min following the addition of 0.4 mM $H_2O_2$ ( + $H_2O_2$). Scale bar, 5 μm. (**g**,**h**) Quantitative time-lapse analysis of the nuclear localization of Tpk1-GFP (**g**) or Tpk2-GFP (**h**) localization in the indicated strains upon the addition of 0.4 mM $H_2O_2$. (**i**) Time to first nuclear localization of Msn2 in the indicated strains expressing the *msn2S288AS304A* mutant allele, bypassing PKA-mediated inhibition of Msn2 nuclear localization upon illumination (115 μW, n = 111 (wt) or 67 (*bcy1Δ*)). Red lines denote median values, boxes 25th and 75th percentiles and whiskers minimum and maximum values of data points not classified as outliers in MATLAB.

PKA catalytic subunits Tpk1 and Tpk2, but not Bcy1 and Tpk3, rapidly redistributed from the nucleus into small cytoplasmic foci (Fig. 4f–h and Supplementary Fig. 4q,r). Such nucleus to cytoplasm redistribution of Tpk1 and Tpk2 in response to $H_2O_2$ depended on Tsa1-mediated signal transduction, as Tpk1 and Tpk2 remained concentrated in the nucleus in cells lacking Tsa1 or carrying a serine substitution of the catalytic Cys48 (Fig. 4g,h). The constitutively cytoplasmic localization of Tpk1 and Tpk2 in cells lacking Bcy1 (Fig. 4f–h) offered a possibility to further test the impact of nucleus to cytoplasm relocalization of PKA catalytic subunits upon the addition of $H_2O_2$ on Msn2 nuclear localization. In cells lacking Bcy1, Msn2 does not localize to the nucleus because of increased PKA-dependent phosphorylation (Fig. 3a). This inhibition can, however, be overcome by substituting Msn2 serines relevant for nuclear transport by alanines (Fig. 3c) and, in fact, Msn2 concentrated much more rapidly in the nucleus of *bcy1Δ msn2S288AS304A* cells (Figs 3c and 4i), suggesting that Bcy1-dependent nuclear concentration of PKA catalytic subunits delays the Msn2 response upon illumination and $H_2O_2$ addition. We conclude that the peroxiredoxin Tsa1 regulates Msn2 at the level of the subcellular localization of the PKA catalytic subunits, and that nuclear concentration of PKA is required for the delayed Msn2 response upon $H_2O_2$.

## Discussion

For some time it has been known that the *S. cerevisiae* transcription factor Msn2 rhythmically shuttles into and out of the nucleus in response to light[6], by a mechanism involving the modulation of PKA-dependent phosphorylation, which prevents Msn2 accumulation in the nucleus[12]. However, it is not known by which mechanism PKA actually detects the presence or absence of light. We have here identified this light-sensing pathway and showed that it comprises a conserved peroxisomal oxidase, Pox1, the antioxidant enzyme peroxiredoxin Tsa1 and its cognate oxidoreductases, the redundant cytosolic thioredoxins Trx1 and Trx2 (Fig. 5). In this pathway, light reduces Pox1, which in turn reduces molecular oxygen, thereby producing $H_2O_2$. $H_2O_2$ subsequently oxidizes Tsa1, which relays its disulfide bond to thioredoxin. The signalling role of the $H_2O_2$ produced by Pox1 is demonstrated by enhancement or inhibition of light signalling upon inactivation or overexpression, respectively, of the $H_2O_2$-scavenging enzyme Ccp1, as well as by the ability of exogenous $H_2O_2$ to induce Msn2 oscillations that mimic those induced by light. Similarly, the importance of the Tsa1-thioredoxin axis as a redox relay in efficient light signal transduction is demonstrated by the requirement for Tsa1 catalytic activity and of thioredoxin oxidation. This interpretation is supported, first, by the loss of light signalling upon mutating Tsa1 catalytic Cys residues, second, by the exacerbation of signal transduction in cells lacking the cytosolic thioredoxin reductase Trr1, which accumulate oxidized thioredoxins and, third, by the loss of increased signalling by further inactivation of the thioredoxins in the Trr1 mutant. The ultimate target of oxidized thioredoxins is yet unknown, as is the nature of their effect on this target. Oxidized thioredoxins could indeed bind to (a) specific target protein(s) and/or they could oxidize Cys residues of this putative target protein(s). Our data, however, show that light signalling via Tsa1 modulates Msn2 nucleo-cytoplasmic shuttling by causing the dephosphorylation of specific PKA-dependent Msn2 serine residues and by

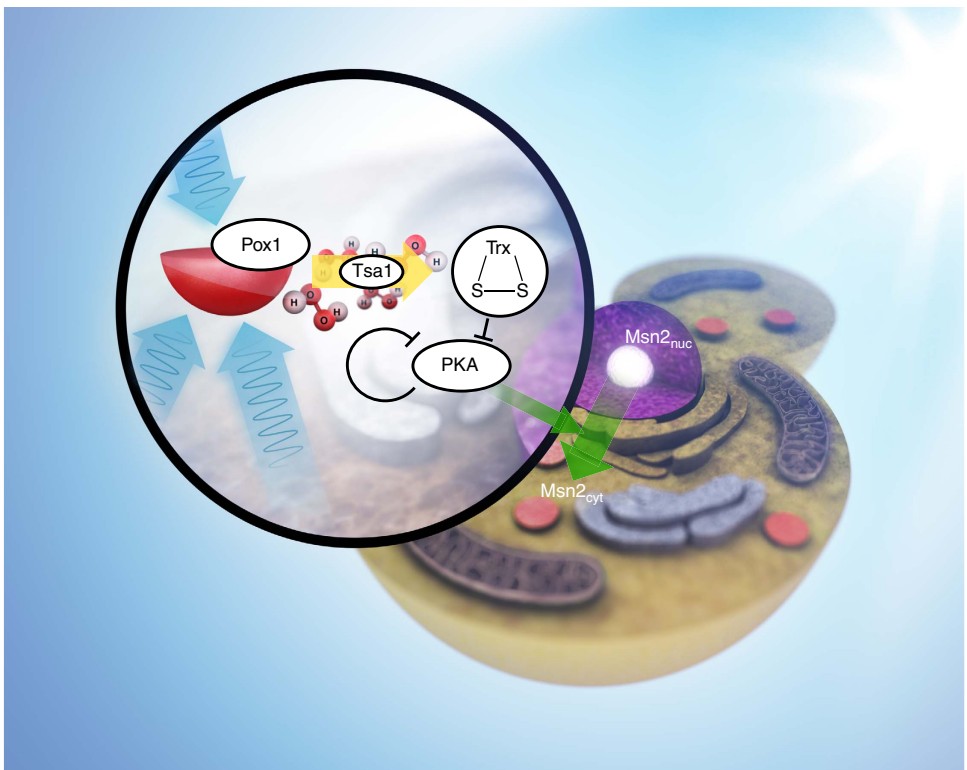

**Figure 5 | Model of light-sensing in the absence of dedicated light receptors and light-induced Msn2 rhythms.** Light reduces flavin-containing oxidases such as peroxisomal acyl-CoA oxidase (Pox1), which produce $H_2O_2$ upon reoxidation by $O_2$. The $H_2O_2$ signal is transduced by the peroxiredoxin Tsa1, which instantly relays the signal to the cytosolic thioredoxins (Trx). Oxidized thioredoxins inhibit PKA-dependent phosphorylation, in part through inhibiting the nuclear retention of the PKA catalytic subunits, allowing Msn2 to concentrate in the nucleus. Negative feedback regulation periodically restores PKA activity, causing Msn2 to exit the nucleus.

counteracting the nuclear retention of the two most highly expressed catalytic subunits of PKA[48]. Tpk2 has previously been shown to be transported out of the nucleus and into small cytoplasmic messenger RNA processing bodies upon nutrient starvation[48] and in cells lacking Igo1 and Igo2 (ref. 49), which regulate the transition to the stationary phase through the $5' \rightarrow 3'$ mRNA decay pathway[50], suggesting that the subcellular localization of PKA is altered also in response to other stimuli that modulate PKA activity. The almost constitutive nuclear localization of Msn2 in a mutant in which neither Tpk1 nor Tpk2 concentrate in the nucleus (bcy1Δ, Figs 3c and 4g,h) and the lack of Msn2 response in mutants not transporting the PKA catalytic subunits out of the nucleus upon the addition of $H_2O_2$ (tsa1Δ and tsa1C48S, Fig. 4g,h), suggests that PKA must exit the nucleus for efficient Msn2 nuclear concentration on $H_2O_2$ addition. In addition, slow Tsa1-mediated nuclear export of the PKA catalytic subunits may contribute to the delayed initial Msn2 nuclear concentration upon illumination or on addition of $H_2O_2$. However, Tsa1 must also regulate PKA dependent phosphorylation at (an) additional level(s), as the nuclear retention of Msn2 is increased in the hyperresponsive tsa1ΔYF mutant upon the addition of $H_2O_2$, whereas neither Tpk1 nor Tpk2 are transported out (Supplementary Fig. 4u,v). Importantly, the Msn2 response was still postponed in tsa1ΔYF mutant cells (Fig. 2d), suggesting that the slow nuclear exclusion of the PKA catalytic subunits is important for the delayed Msn2 nuclear accumulation in response to $H_2O_2$ and light. The additional target(s) of the Tsa1-mediated pathway could be PKA, a regulatory protein in the PKA pathway, or possibly the phosphatases PP1 and PP2A, which also regulate Msn2 nuclear accumulation[10,11]. In fact, PP1 was suggested to modulate glucose starvation-induced Msn2 nuclear localization by targeting PKA-cognate serine residues involved in Msn2 nuclear transport[9].

What is the origin of the rhythmic nature of the oscillations in Msn2 nucleo-cytoplasmic shuttling? One possibility is that the light signal diminishes PKA activity to levels at which negative feedback regulation of the pathway periodically releases upstream stimulatory modules from feedback inhibition, leading to oscillatory Msn2 shuttling into and out of the nucleus[12]. The fact that partial inactivation of PKA activity recapitulated Msn2 oscillations in the absence of light (Fig. 4c,d) point to the importance of sensitive and partial reduction of PKA activity for the oscillatory behaviour. However, a role of the phosphatases cannot be ruled out, based on the fact that deficient Msn2 NES phosphorylation fully restored the lack of light response in cells lacking Tsa1 (Fig. 3c), whereas overproduction of the cAMP phosphodiesterase only partly suppressed this defect (Fig. 4a).

The role of a peroxisomal oxidase in light-induced $H_2O_2$ production in yeast reported here, as well as in mouse, monkey and human cell lines[29], together with the messenger function of $H_2O_2$ in zebrafish, which couples the light signal to the circadian clock[30], indicates that $H_2O_2$ might be a conserved second messenger by which light affects cellular physiology in general and circadian clocks in particular. Interestingly, recent data in flies support a role for $H_2O_2$ in signalling ultraviolet light independent of normal light-sensing pathways through two specific isoforms of the transient receptor potential A1 membrane channel[51]. Our data support a receptor role for peroxiredoxins in the regulation of light-induced ultradian rhythms in yeast and given the high degree of conservation of all components of the pathway, probably also in other organisms. The exquisite sensitivity of peroxiredoxin catalytic cysteines to $H_2O_2$ makes them ideally suited both as receptors to relay light-induced $H_2O_2$ signals, as well $H_2O_2$ scavengers to terminate them[52]. By identifying a peroxiredoxin as a light-receptor of cAMP-PKA, which constitutes a pleiotropic nutrient-responsive pathway regulating, for example, core circadian clock mechanisms[53] and ageing[15], our data contribute to the understanding of the roles of light, $H_2O_2$ and peroxiredoxins in the control of endogenous rhythms in general. The data also suggest that peroxiredoxin hyperoxidation and reactivation play a critical role in modulating endogenous rhythmicity by turning off and on $H_2O_2$-mediated signal transduction, respectively. On a final note, it is well established that aged organisms display less rhythmical circadian clocks[54] and conversely that behaviourally arrhythmic animals exhibit accelerated ageing[55]. Given the roles of peroxiredoxins in both stimulating longevity[21–24] and in controlling biological rhythms[1,25,26], further studies to answer the intriguing question of how the control of these processes are linked through peroxiredoxins are warranted.

## Methods

**Strains.** The strains used in this study are listed in Supplementary Table 1. Strain YMM141 was constructed by amplifying trr1Δ::kanMX4 from genomic DNA from a YPH98 trr1Δ::kanMX4 strain[56] and introducing it into strain BY4743 (ref. 57), selecting for G418 resistance and verifying the deletion by diagnostic PCR. The heterozygous diploid was sporulated and a resulting Mat alpha, Met[+], Lys[−] and G418[+] spore constitutes YMM141. To construct strain YMM142, a Mat a, Met[+], Lys[−], trr1Δ::kanMX4 spore obtained in the above-described sporulation that generated YMM141 was crossed to YMM114. A resulting Mat alpha, Met[+], Lys[−], Nat[+] and G418[+] spore constitutes strain YMM142. To construct strains YMM143 and YMM144, trx1Δ::hphMX4 was amplified from genomic DNA from a BY4742 trx1Δ::hphMX4 strain (trx1Δ::kanMX4 from the deletion collection Research Genetics[58], marker switched[59] to trx1Δ::hphMX4) and introduced into a BY4742 trx2Δ::natMX4 strain (trx2Δ::kanMX4 from the deletion collection Research Genetics[58], marker switched[59] to trx2Δ::natMX4). The resulting BY4742 trx1Δ::hphMX4 trx2Δ::natMX4 strain was crossed to a Mat a, Met[+], Lys[−], trr1Δ::kanMX4 spore obtained in the above-described sporulation that generated YMM141. The resulting Mat alpha, Met[+], Lys[−], Hyg[+], Nat[+] and G418[−] (YMM143) or G418[+] (YMM144) spores constitute YMM143 and YMM144. Strains YMM145, YMM146, YMM147, YMM148 and YTpkCMx3 were constructed by the delitto perfetto method[60] using plasmid pCORE-UK as a template. Sequencing of PCR products amplified from 5-FOA[R] and G418[−] clones verified the indicated mutations and a sequence otherwise identical to strain BY4742 from 220 bp upstream of the TSA1 open reading frame to 191 bp downstream (YMM145, YMM146, YMM147 and YMM148) or from 296 bp upstream to 81 bp downstream of TPK1, 195 bp upstream to 78 bp downstream of TPK2 and 276 bp upstream to 66 bp downstream of TPK3 (YTpkCMx3). Strain WR175 was constructed by tagging Msn2 with the C-terminal tag HTBeaq coupled to hphMX[49] using standard methods for cassette amplification and integration. Strain YMM149 was constructed by crossing WR175 to strain YMM114. The resulting Mat a, Met[−], Lys[−], Arg[−], Hyg[+], Nat[+] and G418[+] spore constitutes YMM148. Strain YMM150 was constructed by transforming strain WR175 with bcy1Δ::natMX4 amplified from genomic DNA from a BY4742 bcy1Δ::natMX4 strain (bcy1Δ::kanMX4 marker-switched[59] to bcy1Δ::natMX4). A Nat[+] clone displaying reduced glycogen accumulation (as assayed by iodine vapour[61]) was PCR verified and constitutes strain YMM150. Strains YMM160, YMM163, YMM164 and YMM167 were constructed by crossing BY4741 TPK1-GFP and BY4741 TPK2-GFP to BY4742 tsa1Δ::kanMX4 (Research Genetics) and BY4742 bcy1Δ::kanMX4, sporulating the diploid strain generated and selecting Mat a, Met[−], Lys[+], His[+], G418[+] spores. Strains YMM161, YMM162, YMM165 and YMM166 were constructed by crossing strains YMM160 and YMM164 to strains YMM145 and YMM147, sporulating the resulting diploid and selecting for Mat a, Met[-], Lys[+], His[+], G418[−] spores.

**Plasmids.** The plasmids used in this study are listed in Supplementary Table 2. Msn2-GFP was expressed from the plasmid pADH1-MSN2-GFP[8] based on the yCPlac111 backbone containing a LEU2 marker and MSN2 expression controlled by the ADH1 promoter. pHR1668 and pWR78 derive from pAMG5 (ref. 8). The MSN2 sequence encoding amino acids 1–264 was integrated using standard cloning techniques. The S304A point mutations in plasmids pWR76 and pWR78 were constructed using a QuikChange Mutagenesis Kit II (Stratagene) according to the manufacturer's recommendations on template plasmids pAMG and pHR1668, respectively. The plasmid pHyPerRedSc was constructed by synthesizing the HyPerRed $H_2O_2$ sensor[33], codon optimized for yeast (sequence available upon request) and subcloning it into the vector pRS416-GPD[62] using BamHI and EcoRI sites inserted upstream and downstream of the HyPerRedSc open reading frame, respectively (GenScript). C199S point mutagenesis was performed by changing a TGT codon into a TCT in the HyPerRedSc coding sequence (positions 361–363, GenScript). Plasmid pRS315-TRX2-ProteinA was constructed in a two-step procedure. First the TRX2 open reading frame and 300 bp upstream and 164 bp downstream flanked by BamHI and PmlI sites was PCR amplified and cloned into plasmid pRS315 (ref. 8). Second, a Protein A tag was incorporated at the TRX2 C

terminus through PCR amplification of the Protein A tag and a stop codon flanked by AlfI and SnaBI sites using plasmid pBS1479 as a template[63]. Plasmid pWR386 was constructed similarly as plasmid pGGM2-ZF described previously[37], through subcloning of (1) the *GAL4* promoter flanked by SacII and SalI sites PCR amplified from genomic DNA into SacII and SalI sites in plasmid pAMG and (2) a GST-tag digested from plasmid pGEX-5X-3 using SalI–XhoI and subcloned into the SalI site of pAMG. pKP7 was constructed by first subcloning an *MSN2* fragment containing S288A, S582A, S620A, S625A and S633A mutations from pCUP1MSN2A5-GFP[64] into pWR386 and then by introducing the S304A mutation using the QuikChange Mutagenesis Kit II (Stratagene) according to the manufacturer's recommendations.

**Growth conditions.** Strains were grown in the dark at 30 °C in buffered synthetic defined medium (Yeast Nitrogen Base (Formedium) with 0.5% ammonium sulfate, 2% glucose and complete supplement mixture without appropriate amino acids, 1% succinic acid and 0.6% NaOH to buffer the medium to pH 5.8). Overnight cultures were inoculated the day before the experiment and grown to an $OD_{600}$ of 0.4–0.5 in the morning the day of the microscopy experiment.

**Fluorescence microscopy.** Images were acquired using automated epi-fluorescence microscopes (TE2000E-PFS, Nikon Instruments or Olympus IX81) equipped with ×60 oil-immersion objectives (numerical aperture 1.4, Plan Apochromatic, Nikon Instruments or PlanApoN ×60/1.42 Oil, Olympus) and electron-multiplying charge-coupled device cameras (iXon DU-885-CS0-#VP, Andor Technology or a 12-bit Hamatsu camera). The yeast cells were kept in a heated perfusion chamber (FCS2, Bioptechs Inc.) at 28 °C to avoid heat-induced stress responses. The objective was heated to 26.2 °C (according to the manufacturer's instructions) to maintain a stable temperature in the perfusion chamber. The cover glasses was precoated for 1.5 h with protein concanavalin A, 0.5 µg µl$^{-1}$ in 0.01 M PBS, to immobilize yeast cells on the surface. The coating did not induce any stress response. By combining neutral density filters, the illumination intensity was in most experiments set to 115 µW in the microscope field of view, unless otherwise noted. GFP fluorescence imaging was accomplished with a mercury lamp, an excitation filter (470 ± 20 nm), a dichroic mirror (505 nm) and an emission filter (515 ± 15 nm) (Chroma Technology) using the Nikon microscope. On the Olympus microscope, GFP was imaged using a quadruple emission filter at 521 ± 17 nm and an excitation bandpass filter at 485 ± 10 nm. The microscope setups were controlled with the NIS-Elements software (Nikon) and the Xcellence software (Olympus), respectively. HyPerRedSc fluorescent imaging was accomplished in the Olympus system using a quadruple filter set with 606 ± 25 nm for detection and an excitation bandpass filter at 560 ± 12.5 nm was used for imaging.

**Illumination of cells.** Illumination of cells was accomplished simultaneously with GFP excitation as previously described[14]. Cells were continuously illuminated with blue light (450–490 nm) and fluorescence images were typically acquired every 4 s. Bright-field images acquired once every 60 s with a small focus offset were used to identify cells in the CellStat software[14]. After each Msn2-GFP imaging experiment, non-illuminated cells were imaged to ensure that only cells that had been exposed to light exhibited nuclear Msn2p localization.

**$H_2O_2$ treatment.** Growth medium containing $H_2O_2$ at the desired concentration was injected into the perfusion chamber at the start of the experiment. Bright-field and fluorescence images were captured every 30 s at a low intensity (26 µW) to avoid light-induced stress (Supplementary Fig. 1c). The $H_2O_2$ solution was made fresh before the start of each experiment.

**Image and signal analysis.** Image and signal analysis was performed using the CellStat software[14]. To quantify the degree of Msn2 nuclear localization, the signal intensity in the nucleus was compared with that in the cytosol[14]. Signal analysis was implemented using routine MATLAB commands. The nucleocytoplasmic localization trajectories were ordered in MATLAB by means of non-metric multidimensional scaling from low responders to high responders[65].

**Cytosolic $H_2O_2$ determination by HyPerRed.** Median HyPerRedSc and HyPerRedScC199S fluorescence values for each cell and time point was exported from CellStat following cell recognition as for Msn2 signal analysis. Using these fluorescence values, cytosolic $H_2O_2$ was measured by normalizing $H_2O_2$- and bleaching-sensitive HyPerRedSc fluorescence in each cell (Supplementary Fig. 1b,c,f), first to fluorescence values before light/$H_2O_2$ treatment[32] and, second, to the average $H_2O_2$-insensitive (Supplementary Fig. 1f) and bleaching-sensitive HyPerRedScC199S fluorescence (Supplementary Fig. 1b,c,f) at the corresponding time point. Graphs presented represent the average bleaching-corrected HyPerRed fluorescence (Fig. 1c,h and Supplementary Fig. 1g).

**Ccp1 immunoblot analysis.** Levels of Ccp1 protein were determined using immunoblot analysis as previously described[22], the Odyssey Infrared Imaging system (LICOR Biosciences) and rabbit anti-Ccp1 serum[66] (dilution 1:4,000) kindly provided Dr T. Tatsuta. Simultaneous detection of the Pgk1 protein using a mouse anti-Phosphoglycerate Kinase Monoclonal Antibody (22C5D8, Novex part no 459250, Thermo Scientific, dilution 1:10,000) served as a loading control.

**Two-dimensional PAGE analysis.** Two-dimensional PAGE analysis to assay Tsa1 sulfinylation was performed using pre-cast 18 cm nonlinear pH 3–10 IPG strips (Immobiline DryStrip, number 17-1235-01, GE Healthcare, Uppsala, Sweden) in the first dimension, 13.5% Duracryl (Millipore) polyacrylamide gels as the second dimension and silver staining to detect total protein[22].

**Thioredoxin redox-state analysis.** The redox state of thioredoxins was assessed by immunoblot analysis of protein extracts treated with 4-acetamido-4′-maleimidylstilbene-2,2′-disulfonic acid from cells lysed in Trichloroacetic acid as described[48]. Proteins were separated on 20% SDS–PAGE gels. Protein-A-tagged Trx2 expressed from plasmid pRS315-*TRX2*-ProtA was detected by an IgG antibody (from rabbit serum, Sigma Aldrich catalogue number I5006, final concentration 1 µg ml$^{-1}$ or 1:10,000 dilution of a 10 mg ml$^{-1}$ stock solution) in Fig. 2e,f and in Supplementary Fig. 2l–n, whereas endogenous Trx1/2 were analysed by an anti-Trx1/2 antibody[42] (rabbit polyclonal, dilution 1:1,000) in Fig. 2h and Supplementary Figs 2o,4e–h as described[42]. Uncropped pictures of a representative redox-immunoblot analysis of Trx2-ProteinA is shown in Supplementary Fig 2l and of anti-Trx1/2 redox-immunoblot analysis in Supplementary Fig. 2o.

**Mass spectrometry.** SILAC, protein purification via HTBeaq tandem affinity tagging, preparation of samples for MS, tryptic digestion (recombinant, proteomics grade, Roche) and phosphopeptide enrichment via TiO$_2$-tips (Glygen) were performed as described elsewhere[44,67].

Ratios ($H_2O_2$-treated/untreated) of Msn2 phosphopeptides were obtained by comparing the MS1 peak intensities of SILAC-labelled peptides from $H_2O_2$-treated cells grown in the presence of a light-isotope reagent ($^{12}$C-arginine and -lysine) with the peak intensities of untreated cells labelled with a heavy-isotope reagent ($^{13}$C-arginine and -lysine). To quantify phosphorylation in untreated *tsa1Δ* cells (Supplementary Tables 1 and 2), peak intensities were compared between *tsa1Δ* cells grown in the presence of a light-isotope reagent ($^{12}$C-arginine or -lysine) and wild-type cells labelled with a heavy-isotope reagent ($^{13}$C-arginine or -lysine).

Samples were subjected to reversed-phase nano-HPLC (Ultimate 3000, Dionex, Sunnyvale, CA, USA) using the following setup: peptides were loaded onto a precolumn (PepMAP C18, 0.3 × 5 mm, Dionex), desalted for 30 min with 0.1% trifluoroacetic acid (20 µl min$^{-1}$), and eluted onto an analytical column (PepMAP C18, 75 µm × 150 mm, Dionex) for 1 h by means of an organic solvent gradient (275 µl min$^{-1}$), as described elsewhere[67]. The nano-HPLC was directly coupled to an LTQ Orbitrap Velos mass spectrometer (Thermo Scientific) via a nanoelectrospray ion source (Proxeon). The capillary temperature was set to 200 °C. The source voltage on the metal-coated nanoelectrospray ion emitters (New Objective) was 1.5 kV. The 12 most abundant ions were subjected to MS/MS analysis and monoisotopic precursor selection was enabled. For some of the experiments (indicated as set I in the material available online at proteomeXchange.org, see below), MS3 analysis was performed; for the other experiments (set II), multistage activation was triggered at neutral losses of 98, 49 and 32.6. Fragmented precursors were excluded from further fragmentation for 30 (respect. 60) s (with 5 p.p.m. accuracy) and singly charged peptides were generally excluded from MS/MS analysis.

Peptide identification and SILAC quantification were performed using the SEQUEST algorithm in the Proteome Discoverer software package (ver. 1.4.0.288, Thermo Fisher Scientific) with settings similar to those described elsewhere[67]. In short, phosphorylation of Ser/Thr/Tyr, neutral loss of water from Ser/Thr, oxidation of Met and $^{13}$C$_6$ Lys/Arg were set as variable modifications. In addition, carbamidomethylation of Cys was set as a static modification. Spectra were searched against the *Saccharomyces* Genomic Database (6717 entries, 03 February 2011) including contaminants with tryptic specificity, allowing two missed cleavages, a precursor mass tolerance of 3 p.p.m. and a fragment ion tolerance of 0.8 Da. Quantification settings were the default values for the precursor ion quantifier and the event detector was set to 2 p.p.m. The results were filtered at the XCorr values to a false discovery rate of 1% on the peptide level, as described elsewhere[68]. Additional manual validation of peak assignments was performed in the Xcalibur Qual Browser (Thermo Scientific) using a 2 p.p.m. mass window centred on the exact precursor mass. Light/heavy (L/H) ratios of each SILAC experiment were normalized and Arg-Pro conversion was taken into account as described elsewhere[44,67].

The probability of phosphorylation site localization was calculated using phosphoRS 2.0 software[69] implemented in Proteome Discoverer. A phosphorylation site probability of 75% or higher was considered confidently localized. All identified peptide spectrum matches of Msn2 can be found in Supplementary Data 2. All raw MS data files have been deposited at the ProteomeXchange Consortium[70] (http://www.proteomexchange.org) via the PRoteomics IDEntifications partner repository with the dataset identifier PXD001853. Peptide entries in Supplementary Data 2 were filtered for the presence of an L/H ratio and phosphopeptides with a confidence in phosphorylation site allocation other than 'high' (high = phosphoRS 2.0 probability of ≥75%) were

removed. The remaining peptide entries were grouped according to their biological samples (all.raw files of the same sample) and designated as replicates of an experimental condition. Phosphopeptides spanning the same set of phosphorylated residues were further grouped into and designated as a phosphorylation site group (PSG). Unphosphorylated counter peptides (amino acid sequence must harbour at least one of the phosphorylated amino acid residues of the corresponding PSG) were additionally grouped. PSG and counter-peptide ratios were $\log_2$ transformed and averaged, and a confidence interval of $\pm 1$ s.d. was calculated. Resulting $\log_2$ values were reversed and are summarized in Supplementary Data 1 (see worksheets 'n = 2 (biological replicates)' (hidden columns) and 'n = 1 (biological replicates)'). $\log_2$ averaged ratios of all individual biological replicates of an experimental condition were further used to calculate the average ratio and confidence interval of the corresponding experimental condition. The resulting $\log_2$ values were reversed and are summarized in Supplementary Data 1.

**Immunoprecipitation of Msn2 and NES phosphorylation analyses.** Cells were harvested using filtration at the indicated time points following the addition of $H_2O_2$. Cells were lysed using 10 mM Tris-HCl (pH 7.5), 150 mM NaCl, 0.5 mM EDTA, 0.5% Triton X-100, protease inhibitor cocktail (Roche), 1 mM sodium vanadate and 1 mM NaF. Cell lysates were incubated for 1 h at 4 °C with anti-GFP beads (GFP-Trap_A, Chromotek). Beads were washed with lysis buffer and eluted in loading buffer. Total amounts of the Msn2NES-GFP fusion protein expressed from pAMG9 (ref. 8) were visualized using anti-GFP (Roche Molecular Biochemicals, dilution 1:10,000) and PKA-dependent phosphorylation using a phospho-PKA substrate antibody (number 9624, Cell Signaling, dilution 1:1,000). A representative uncropped blot picture of the anti-GFP and anti-phospho PKA immunoblot analysis of Msn2-NES phosphorylation can be seen in Supplementary Fig. 3n. The detection of PKA-specific phosphorylation by the phospho-PKA substrate antibody was verified by complete loss of antibody reactivity on total yeast extracts prepared from the strain YTpkCMx3 upon treatment with 500 nM 1-nM-PP1.

**Statistical analyses.** To assess statistically significant Msn2 localization differences in mutant strains and differences in various treatments, analysis of variance and Tukey's tests were performed on data obtained from the 685 time points between 10 and 60 min following the start of the treatment. Analysis of variance confirmed similar variance between groups compared. Tukey's tests were suitable, because such Msn2 localization data mostly conformed well to a normal distribution. A P-value of <0.05 was considered to indicate a statistically significant difference and, for all experiments, percentages of data points expressing significant differences were calculated (see Supplementary Note 1). Values for the time Msn2 spent in the nucleus or the time to first Msn2 nuclear localization were less clearly normally distributed and were instead tested for statistical significance using two-sided Mann–Whitney U-test[14] (see Supplementary Note 1).

To assess statistically significant differences between HyPerRed fluorescence in different strains or between HyPerRed and HyPerRedC199S expressing strains, non-bleaching corrected fluorescence values were compared (Supplementary Fig. 1b,c,f) using two-sided Mann–Whitney U-tests (see Supplementary Note 1). This test was chosen because the distribution of HyPerRedSc or HyPerRedScC199S fluorescence was skewed towards highly fluorescing cells.

Differences in the amount of Msn2-NES phosphorylation (Fig. 3h) were analysed for statistical significance using two-sided Student's T-tests (see Supplementary Note 1).

**Data availability.** All raw MS data files have been deposited at the ProteomeXchangeConsortium (http://proteomexchange.org) via the Proteomics IDEntifications partner repository with the dataset identifier PXD001853. The data that support the findings of this study are available from the corresponding author on request. The MATLAB code used in signal analysis is available from the corresponding author upon request.

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

## Acknowledgements

Christoph Schüller and Thomas Nyström are acknowledged for comments on the manuscript and we thank Anders Blomberg and Gustav Ammerer for support throughout this study. We are also grateful to Dorothea Anrather and David Hollenstein for help with MS analyses; Marija Cvijovic for help with statistical analyses; Frederik Eisele, Takahashi Tatsuta, Thomas Langer, Joseph Heitman and Joakim Norbeck for providing reagents, and the Centre of Cellular Imaging, Sahlgrenska Academy, for the use of imaging equipment and support from the staff. The Olympus Cell^R microscope was purchased with support from Ingabritt och Arne Lundbergs stiftelse. This work was supported by grants from the Swedish Research Council (to M.M.), by the foundations of Olle Engkvist Byggmästare and Carl Trygger (to M.M.), by the Knut and Alice Wallenberg foundation (to M.K.) and by ANR ERRed and PLBIO INCA_5869 (to M.B.T.).

## Author contributions

M.M., M.K., K.B., K.P., F.R. and W.R. conceived and designed the experiments. M.M. and M.B.T. wrote the manuscript. All authors commented and edited the text. K.B., K.P., F.R., N.R., A.I., G.P. and W.R. performed experiments and analysed data. N.W. analysed data.

## Additional information

**Competing interests:** The authors declare no competing financial interests.

**Publisher's note**: 

