## [Peer Review File · Nature Communications]

Reviewers' Comments:

Reviewer #1 (Remarks to the Author):

General comments:

This represents an interesting study in which the authors have attempted to demonstrate the shuttling of Msn2 between the nucleus and cytosol in response to light stimuli or treatment with the hydrogen peroxide in yeast. They demonstrate that H₂O₂ generated by Pox1 upon the light stimulus is sensed by the peroxiredoxin Tsa1 and further transduced to thioredoxin to inhibit PKA activity. The authors should be cautious in extension of their findings to circadian rhythms (e.g. in the Abstract), which is of course speculative, and not shown directly by this study.

Major points:

1. The authors make a statement in the abstract 'In particular, the involvement of H₂O₂ as a second messenger in signaling links Msn2 oscillations to light-induced entrainment of circadian rhythms and suggest conserved roles for peroxiredoxins in endogenous rhythms' No substantiate data is presented demonstrating the light-induced entrainment of circadian rhythms in yeast. They should remove this from the abstract and leave mention to the discussion, where they present their findings in the correct context already.
2. The authors did not demonstrate the accumulation of Msn2 in the dark phase, which would be needed to state it as 'an entrainable rhythm'. To show this, an experiment during a light-dark dark cycle would need to be performed. Similarly, to state findings as 'circadian', a free-running experiment (under constant conditions) would need to be done.
3. What happens to Msn2 localization with longer light exposure? When do they reach closer to ~ 100% nuclear localization? Figure 4c, shows with the higher concentrations of H₂O₂ nuclear localization of Mns2 is near 100%. Is this effect seen in tsa1Δ or tsa1C48S mutants?
4. The authors have plotted all the heatmaps without clustering and this make it difficult for readers to comprehend. Could the authors plot clustered heatmaps to help discern patterns? The clustered heatmaps may give more of visual information that might change interpretations.
5. Figure 1d,1f: why are the wild types behaving differently? In Fig 1d, Msn2 localisation plateaus at 20% at 20 min, but in Fig 1e localization rises progressively to reach a maximum (30%) at 60 min. The findings would appear inconsistent in the wild type cells used.
6. What happens if PDE2 is over expressed in the tsa1C48S? Do you see any oxidation of thioredoxins in tsa1C48S mutants upon exposure to light or H₂O₂?

7. Why have the authors used *tsa1Δ* rather than *tsa1C48S*? The latter mutant strain had least response to the light stimuli when compared to *tsa1Δ*, and in the deletion strain, *tsa1Δ*, there could be a compensation mechanism acting, and hence they show a response to the light in nuclear localization of Msn2.

Minor points:

1. It would be helpful to the readers if the authors could present the number of replicates, error bars and the statistical test outcomes on the graphs. Most of the line graphs are missing the error bars and it is not clear how many times these experiments were done and shown to be representative.

2. Figure 1b, could the authors present the mean/median of the nuclear localization in the population instead of a single cell. Could the authors comment on the dotted line in the graph?

3. Line 274: 'PKA is required for the delayed Msn2 response upon illumination and H₂O₂' - no data appear to be shown for light stimulation and H₂O₂.

Reviewer #2 (Remarks to the Author):

The authors describe a mechanism by which a signal from blue light is transduced within the cell to regulate the localization of the stress-responsive transcription factor Msn2. This work, and that of others, reveals a scenario in which light and oxidative species are similar in their effects on the cell and is of general interest in providing a mechanistic explanation for how cellular processes within simple organisms, which lack dedicated photoreceptors, might be coordinated in response to light. It also reveals how signal transduction can be modulated downstream of the activating signal and adds to the growing number of roles associated with the conserved family of peroxiredoxins. This is a substantial body of work that has been carefully conducted, represents the data well, in a number of formats, and concludes with a model that summarizes the findings.

Despite this, I would like to see the major claim upon which the mechanism is based - namely that peroxisomal oxidase, Pox1, converts light into a hydrogen peroxide signal made stronger, by including data showing that hydrogen peroxide is generated in response to light. The conclusion is likely given that the readout (Msn2 translocation into the nucleus) is similar for a peroxide or light stimulus, is reduced in a strain deleted for Pox1 and is greater in a strain compromised for peroxide scavenging, however, the readout in each case is Msn2 localization and the presence of peroxide/reactive oxygen has not been examined. A figure showing that 115μW light and 0.4mM peroxide result in similar levels of ROS and that this effect is reduced or abrogated in a strain deleted for POX1 would substantially strengthen this point.

Minor points/typos

1. Supp. Fig.11, 'e' missing in 'wild type'

2. Figure 2g, why is the trajectory of nuclear localization of Msn2 in WT in response to light different from that in other experiments? ~40% cells have nuclear Msn2 at 60' post illumination cf. ~20% in most other experiments.
3. Line 201 states 'However, the Msn2 response to high levels of H₂O₂ (0.8mM) was independent of protein kinase (PKA) activity (Supplementary Figure 3b)'. In the figure the response to 0.8mM H₂O₂ appears to be sensitive to PKA and the response to 0.4mM is not. There is an inconsistency here.
4. Supp. Methods yMM145 cf. YMM145

Response letter Reviewer's comments NCOMMS-16-08587A-Z

'Light-sensing via hydrogen peroxide and a peroxiredoxin'

Reviewer #1 (Remarks to the Author):

General comments:

This represents an interesting study in which the authors have attempted to demonstrate the shuttling of Msn2 between the nucleus and cytosol in response to light stimuli or treatment with the hydrogen peroxide in yeast. They demonstrate that H₂O₂ generated by Pox1 upon the light stimulus is sensed by the peroxiredoxin Tsa1 and further transduced to thioredoxin to inhibit PKA activity. The authors should be cautious in extension of their findings to circadian rhythms (e.g. in the Abstract), which is of course speculative, and not shown directly by this study.

Major points:

1. The authors make a statement in the abstract 'In particular, the involvement of H₂O₂ as a second messenger in signaling links Msn2 oscillations to light-induced entrainment of circadian rhythms and suggest conserved roles for peroxiredoxins in endogenous rhythms' No substantiate data is presented demonstrating the light-induced entrainment of circadian rhythms in yeast. They should remove this from the abstract and leave mention to the discussion, where they present their findings in the correct context already.

We agree that the way we stated the connection between Msn2 oscillations and light-induced entrainment of circadian rhythms in the abstract previously was a little unfortunate and might be interpreted as if we present data directly linking the two. In the revised version of the ms we have re-written this sentence to clarify what we mean, namely that H₂O₂ acts as a second messenger in both processes.

Abstract : 'In particular, the use of H₂O₂ as a second messenger in signaling is common to both Msn2 oscillations and light-induced entrainment of circadian rhythms, which suggests conserved roles for peroxiredoxins in endogenous rhythms.'

2. The authors did not demonstrate the accumulation of Msn2 in the dark phase, which would be needed to state it as 'an entrainable rhythm'. To show this, an experiment during a light-dark dark cycle would need to be performed. Similarly, to state findings as 'circadian', a free-running experiment (under constant conditions) would need to be done.

We do not think that the Msn2 rhythm presented in this paper constitutes an entrainable mechanism. Therefore, although they might give interesting results, we strongly believe that experiments on Msn2 dark phase accumulation and free-running is beyond the scope of this paper.

3. What happens to Msn2 localization with longer light exposure? When do they reach closer to ~ 100%

nuclear localization? Figure 4c, shows with the higher concentrations of H₂O₂ nuclear localization of Msn2 is near 100%. Is this effect seen in tsa1Δ or tsa1C48S mutants?

Here we are a bit unsure what the reviewer means since Fig. 4c depicts Msn2 localization not upon H₂O₂ treatment but upon high levels of PKA inhibition achieved by a chemical genetic strategy. In fact, such a dramatically increased nuclear localization as upon the addition of 500 nM of the engineered PKA inhibitor 1-NM-PP1 is not observed upon treatment with H₂O₂ or illumination.

Nevertheless, our previous studies indicate that Msn2 nuclear localization does increase with the accumulated light dose, which is a function of both the time of light exposure and the intensity of the light (Bodvard K et al BBA 2011, Logg K et al FEMS Yeast Research 2009). In fact, upon illuminating cells with an increased light dose (18.5 Jcm⁻²s⁻¹, about 40-fold higher than most of the experiments performed in this paper) administered mostly by increasing light intensity (Logg K et al, 2009), about 70% of cells accumulate Msn2 in the nucleus. Importantly, the Msn2 response to such a high light dose is of a similar magnitude as upon the addition of increased levels of H₂O₂ (e.g. 1 mM, Fig. S1h, j), further reinforcing the similarities between the light- and H₂O₂-responses.

To answer the question if Tsa1 is necessary also for the responses to increased H₂O₂ levels/increased light intensity we addressed Msn2 nuclear localization upon illumination at 200 μW and following the addition of 0.8 or 1 mM H₂O₂ in the revised version of the manuscript. We found that Msn2 concentrated in the nucleus only upon the addition of 1 mM H₂O₂ suggesting that Tsa1 is needed for cells to respond also to H₂O₂/light stress of increased intensity but that it is possible to overcome the requirement for Tsa1 at extreme stress. These new data are discussed in the text (p6 lines 15-20, see below) and clarify the physiological importance and the limits of the reported light and H₂O₂ sensing mechanism.

‘However, the Msn2 response to high levels of H₂O₂ (0.8 mM) was not inhibited by increased protein kinase A (PKA) activity (Supplementary Fig. 3b). On the contrary and in agreement with previously published data⁴¹, the response to high levels of H₂O₂ was stimulated in cells expressing constitutive PKA activity, indicating that (an)other pathway(s) operates under this condition. At these levels of H₂O₂, Tsa1 was still required for the Msn2 response⁴¹ (Supplementary Fig. 3c) whereas upon further increased H₂O₂ (1 mM), it was not (Supplementary Fig. 3c), suggesting that the requirement for Tsa1 could be overcome upon increased H₂O₂.‘

4. The authors have plotted all the heatmaps without clustering and this make it difficult for readers to comprehend. Could the authors plot clustered heatmaps to help discern patterns? The clustered heatmaps may give more of visual information that might change interpretations.

We thank the reviewer for this excellent suggestion and are happy to now present all color maps clustered by a non-metric multidimensional scaling method to order cells from low to high responders. This new method we believe facilitates visualization, since e.g. the lack of Msn2 response in Tsa1-deficient cells becomes even more evident, a weak remaining response seen only in a few cells (Supplementary Fig. 3e).

5. Figure 1d,1f: why are the wild types behaving differently? In Fig 1d, Msn2 localisation plateaus at 20% at 20 min, but in Fig 1e localization rises progressively to reach a maximum (30%) at 60 min. The findings

would appear inconsistent in the wild type cells used.

In these two experiments, the wild-type cells used are different and so is the growth medium used. Whereas the wild-type in Fig 1d is *TRP1* prototrophic and grows in the presence of 50 mg/l tryptophan in the medium, the wild-type in Fig. 1f lacks the chromosomal *TRP1* gene and grows in the absence of tryptophan supplementation to select for the *TRP1* expressing plasmid pYX132. Nevertheless, we repeated the analysis of Msn2 nuclear localization in the pYX132 containing wt cells (Fig. 1f) and following these additional experiments the response is more similar to the normal wt lacking pYX132 and in fact indistinguishable by statistical analysis (see Supplemental Methods, p18, line s 48-50).

6. What happens if PDE2 is over expressed in the *tsa1C48S*? Do you see any oxidation of thioredoxins in *tsa1C48S* mutants upon exposure to light or H₂O₂?

Given the complete Cys48-dependence of the Tsa1 light- and H₂O₂-sensing (see below) we chose to focus our further experimental analysis on what happens to Trx oxidation in cells lacking Tsa1 altogether upon overproducing *PDE2* to reduce PKA activity. In the revised version, we have added results from experiments assaying the Trx oxidation status upon the addition H₂O₂ in wt and *tsa1Δ* strains o/e or not *PDE2*. Importantly, these results show that overexpression of *PDE2* does not affect the oxidation of Trx upon H₂O₂ addition, neither in wt nor *tsa1Δ* cells. Suppression of the defective Msn2 response in Tsa1 deficient cells is thus rendered through PDE2s well-established role in regulating cAMP levels and PKA. Importantly, these data reinforce the conclusion Tsa1 affects the response to H₂O₂ and light by inhibiting PKA activity. These new data are discussed in the text (p6, line 47-p7, line 4) :

‘In support of the role of Tsa1-dependent light signaling in counteracting PKA-dependent inhibition of Msn2, overexpression of the cAMP phosphodiesterase *PDE2*, which decreases PKA activity⁴⁷, restored the defective light response of the *tsa1Δ* mutant (Fig. 4a, Supplementary Fig. 4a-d), but did not restore the oxidation of thioredoxins (Supplementary Fig. 4g-h). In addition, thioredoxin oxidation was unaffected upon the addition of H₂O₂ in Tsa1-proficient cells overexpressing *PDE2*, further supporting Msn2 signal control downstream of the thioredoxins upon Pde2 overproduction (Supplementary Fig. 4e-f).’

7. Why have the authors used *tsa1Δ* rather than *tsa1C48S*? The latter mutant strain had least response to the light stimuli when compared to *tsa1Δ*, and in the deletion strain, *tsa1Δ*, there could be a compensation mechanism acting, and hence they show a response to the light in nuclear localization of Msn2.

We have performed further experiments assaying Msn2 localization in *tsa1Δ* mutants upon illumination which suggest that such cells hardly respond at all to illumination (see updated Fig 2c). Importantly, with the new data added, the data on Msn2 nuclear localization {‘Nuclear Msn2 (% cells)’}-graphs for *tsa1Δ* and *tsa1C48S* cells are virtually indistinguishable by statistical analysis (see Supplementary Methods, p19, lines 7-9) suggesting that there is no compensatory mechanism at play in *tsa1Δ* cells and that light- and H₂O₂ responses are entirely dependent on Tsa1C48.

Minor points:

1. It would be helpful to the readers if the authors could present the number of replicates, error bars and the statistical test outcomes on the graphs. Most of the line graphs are missing the error bars and it

is not clear how many times these experiments were done and shown to be representative.

In the revised version we have indicated the number of cells (n) used for population analyses in each graph. We have also added information on the number of replicates of each experiment in Supplementary Methods (Supplementary Data, p18, 'Reproducibility and statistical analyses'). Already in the previous version of the manuscript there was a detailed section on the statistical analyses used to evaluate data (Supplementary Data, p18, 'Reproducibility and statistical analyses').

2. Figure 1b, could the authors present the mean/median of the nuclear localization in the population instead of a single cell. Could the authors comment on the dotted line in the graph?

In the revised manuscript we have added a new section outlining the methodology used to study Msn2 localization dynamics upon illumination more in detail (p4, lines 4-11).

'To study this phenomenon we used an experimental setup enabling both single-cell (Fig. 1a–c and Supplementary Movie 1) and population analyses (Fig. 1d) of Msn2 localization dynamics¹⁴. In short, cells were exposed to blue visible light ($\lambda=450-490$ nm) and imaged in a perfusion chamber using epifluorescence microscopy as previously described¹⁴. At a light intensity corresponding to that on a sunny day ($P=115 \mu\text{W}$), Msn2 accumulated into the nucleus and then back to the cytoplasm on average 4.3 ± 3.6 times/h (average \pm SD, $n = 248$) and, following an initial lag of ~ 20 minutes, the proportion of cells exhibiting Msn2 nuclear localization¹⁴ stabilized at around 25% (Fig. 1d).'

By reading this we hope we have made it clear to the readers and the reviewer that the single cell localization data in Fig. 1b is only one of multiple ways that we visualize Msn2 localization data and that it is presented mainly to clarify the experimental procedures used. In fact, the population data requested by the reviewer was present already in the previous version of the manuscript (Fig. 1d, for the wild-type following illumination at $115 \mu\text{W}$) and in all the other figures presenting population data '(Nuclear Msn2 (% cells))'.

The dotted line in Fig. 1b is now explained in the legend to Fig. 1:

'**b.** Single-cell trace representing nuclear localization upon illumination of a wild-type (wt) cell, quantified by comparing the signal intensity in the nucleus with the signal intensity in the cytosol (see Supplementary Methods). The dashed line indicates the threshold ratio of nuclear/cytosolic Msn2 signal for nuclear localization in population analyses¹⁴.'

3. Line 274: 'PKA is required for the delayed Msn2 response upon illumination and H₂O₂' - no data appear to be shown for light stimulation and H₂O₂.

We have rephrased to more accurately describe the experiments that we actually did (p7, lines 41-42):

'PKA is required for the delayed Msn2 response upon H₂O₂).

Reviewer #2 (Remarks to the Author):

The authors describe a mechanism by which a signal from blue light is transduced within the cell to regulate the localization of the stress-responsive transcription factor Msn2. This work, and that of others, reveals a scenario in which light and oxidative species are similar in their effects on the cell and is of general interest in providing a mechanistic explanation for how cellular processes within simple organisms, which lack dedicated photoreceptors, might be coordinated in response to light. It also reveals how signal transduction can be modulated downstream of the activating signal and adds to the growing number of roles associated with the conserved family of peroxiredoxins. This is a substantial body of work that has been carefully conducted, represents the data well, in a number of formats, and concludes with a model that summarizes the findings.

We thank the reviewer for an excellent summary of our work and the encouraging feed-back.

Despite this, I would like to see the major claim upon which the mechanism is based - namely that peroxisomal oxidase, Pox1, converts light into a hydrogen peroxide signal made stronger, by including data showing that hydrogen peroxide is generated in response to light. The conclusion is likely given that the readout (Msn2 translocation into the nucleus) is similar for a peroxide or light stimulus, is reduced in a strain deleted for Pox1 and is greater in a strain compromised for peroxide scavenging, however, the readout in each case is Msn2 localization and the presence of peroxide/reactive oxygen has not been examined. A figure showing that 115 μ W light and 0.4mM peroxide result in similar levels of ROS and that this effect is reduced or abrogated in a strain deleted for POX1 would substantially strengthen this point.

This is an excellent suggestion and we are happy to present data on Pox1-dependent H₂O₂ generation in the revised version of the manuscript (see p4, lines 17-27, Fig. 1e, h and Supplementary Fig. 1b, c, g).

‘To substantiate this, we used the cytosolic genetically encoded H₂O₂ sensor HyPerRed³⁴, which we reasoned would be less sensitive to bleaching upon illumination with blue light than GFP-based H₂O₂ probes (e.g. HyPer³⁵ and roGFP2-Tsa2 Δ CR³⁶). Upon the start of illumination the HyPerRed signal initially stayed constant but already after a couple of minutes decreased steadily at a rate dependent on light intensity (Supplementary Fig. 1b-c), indicating that probe bleaching interfered with its use in H₂O₂ determination. Nevertheless, in cells expressing an H₂O₂-insensitive probe (carrying a C199S substitution in the OxyR H₂O₂-sensing domain³⁴), fluorescence decreased faster and/or to lower levels than in cells expressing the H₂O₂-sensitive probe (Supplementary Fig. 1b-c) indicating that upon illumination the levels of cytosolic H₂O₂ increased (Fig. 1e). The Cys199-dependent HyPerRed fluorescence increased as a function of light intensity (Fig. 1e) suggesting that so do the levels of cytosolic H₂O₂.’

We would like to point out, however, that measurements of H₂O₂ using modern, sensitive and specific genetic probes (e.g. HyPer and roGFP2-Tsa2 Δ CR) upon illumination are hampered by the fact that the light stimulus in this study consists of blue light (i.e. practically the same wavelengths as those used to image the probes). Nevertheless, we bypassed this technical hurdle by imaging both an H₂O₂-sensitive HyPerRed probe and its H₂O₂-insensitive mutant, lacking the Cys199 of the OxyR sensor domain (see Supplementary Figs. 1b,c). Importantly, the measurement of fluorescence using both these probes and in wt and Pox1-deficient cells demonstrate a Pox1-dependent increase in cytosolic H₂O₂ (measured as C199-dependent HyPerRed fluorescence, Fig 1e,h) which is comparable to the increase seen upon the addition of 0.4 mM exogenous H₂O₂ (Supplementary Fig. 1g). As foreseen by the reviewer this new data substantially strengthens the key point of blue light inducing a Pox1-dependent H₂O₂-signal of the paper.

Minor points/typos

1. Supp. Fig.1l, 'e' missing in 'wild type'

Typo corrected in the revised version of the paper.

2. Figure 2g, why is the trajectory of nuclear localization of Msn2 in WT in response to light different from that in other experiments? ~40% cells have nuclear Msn2 at 60' post illumination cf. ~20% in most other experiments.

We have replaced the old wild-type curve in Fig. 2g, that was acquired using fewer cells and lingered in the figure from an old version, with one analyzing Msn2 nuclear localization in more cells, the same one used throughout the study.

3. Line 201 states 'However, the Msn2 response to high levels of H₂O₂ (0.8mM) was independent of protein kinase (PKA) activity (Supplementary Figure 3b)'. In the figure the response to 0.8mM H₂O₂ appears to be sensitive to PKA and the response to 0.4mM is not. There is an inconsistency here.

We believe that the different pathways that operate upon low levels of H₂O₂, which can be argued to be more relevant to conditions upon illumination, and high levels has made the reviewer a bit confused. In the revised version of the ms we have expanded the section describing the different pathways in operation at different levels of H₂O₂ and also added the results of further experiments (p6, lines 13-23, Supplementary Fig. 3c).

Light-induced Msn2 nuclear accumulation was prevented in cells with constitutive PKA activity (*bcy1Δ* or *pde2Δ*, Fig. 3a), as shown previously^{12,14}, and, importantly, the same was true for H₂O₂ at levels that recapitulated the light response (0.4 mM, Supplementary Fig. 3a-b). However, the Msn2 response to high levels of H₂O₂ (0.8 mM) was not inhibited by increased protein kinase A (PKA) activity (Supplementary Fig. 3b). On the contrary and in agreement with previously published data⁴¹, the response to high levels of H₂O₂ was stimulated in cells expressing constitutive PKA activity, indicating that (an)other pathway(s) operates under this condition. At these levels of H₂O₂, Tsa1 was still required for the Msn2 response⁴¹ (Supplementary Fig. 3c) whereas upon further increased H₂O₂ (1 mM), it was not (Supplementary Fig. 3c), suggesting that the requirement for Tsa1 could be overcome upon increased H₂O₂.

Already in the previous version of the ms our data clearly supported an Msn2 response to lower levels of H₂O₂ antagonized by PKA (Supplementary Fig 3a-b) and we believe that we have made this important distinction between responses to low and high H₂O₂ more clear in the revised version. As outlined in the response to point 3, reviewer 1 these data address both the physiological importance and the boundaries of the here discovered sensing mechanism and support a critical role for Tsa1 in light and H₂O₂ detection at physiological levels.

4. Supp. Methods yMM145 cf. YMM145

Typo corrected in the revised version of the manuscript

Reviewers' Comments:

Reviewer #1 (Remarks to the Author):

The authors have addressed the major concerns raised in the previous round of review, by both new experiments and by clarifications to the text. Therefore, the paper is now appropriate for publication.

Reviewer #2 (Remarks to the Author):

The authors have now added data to show that both light and addition of hydrogen peroxide result in an increase in intracellular peroxide and have made a number of revisions that address the reproducibility of their findings. These changes substantially strengthen the manuscript.

Although most of the figures show the same wild type, the wild type in Figure 1g is different and the percentage of cells with nuclear Msn2 is about 1.5 times greater. This doesn't change the findings, as the point is to illustrate the difference between the wild type and the mutant, however, the result should be commented on.

Reviewer #3 (Remarks to the Author):

I was asked to specifically look at the proteomic data in the paper.

The data search and phospho site assignment is fine I think (PD, SEQUEST and phosphoRS). These were done by standard methods.

However, there are issues with the reporting of the data and also with replicates. The data tables in the supplement are confusing at best; The organization of the MS supplemental data makes it pretty impossible to do anything but take their word for it (data from several different methods and conditions are combined to provide evidence for each specific site, and site specific quantitation is separated from phosphosite evidence - meaning we don't know which PSM they are quantifying off of). They only have one replicate at best two for some conditions and I'm not sure why. If one replicate didn't work you can't just remove it for some site data and not remove it for some other site data, as appears to be the case. For "0.4mM H₂O₂ 10min" in Table S2 it is even written replicate 1 and replicate 3... what happened to replicate 2? Once they know which site(s) they are interested in, they could have re-run these samples with an inclusion list or some other method to make sure they detect it in every replicate.

The standard for "site quantification" would be triplicates and the data as it stands doesn't really reach that standard in some cases. Minimally they would need to say in those cases that the data is from a single experiment.

Reviewers' comments:

Reviewer #1 (Remarks to the Author):

The authors have addressed the major concerns raised in the previous round of review, by both new experiments and by clarifications to the text. Therefore, the paper is now appropriate for publication.

We thank the reviewer for these encouraging words and for constructive feed-back and are delighted to hear that the paper is now appropriate for publication.

Reviewer #2 (Remarks to the Author):

The authors have now added data to show that both light and addition of hydrogen peroxide result in an increase in intracellular peroxide and have made a number of revisions that address the reproducibility of their findings. These changes substantially strengthen the manuscript.

Although most of the figures show the same wild type, the wild type in Figure 1g is different and the percentage of cells with nuclear Msn2 is about 1.5 times greater. This doesn't change the findings, as the point is to illustrate the difference between the wild type and the mutant, however, the result should be commented on.

We are not sure on what grounds the reviewer claims that the control wild-type cells in Fig. 1g show 1.5x increased Msn2 nuclear localization compared to the wild-type cells used eg in Fig. 1d and suspect a misunderstanding (see the figure below). When the 'standard' wild type and the wild type containing plasmid pYX132 are plotted in the same graph the curves overlay well. In fact, upon statistical analysis there is no significant difference between the two strains at any of the 685 time-points sampled. This was pointed out in the previous Rebuttal letter and in Supplementary Information p18, lines 5-7:

‘The response in wt control cells (carrying plasmid pYX132) does not differ significantly from the wt without the plasmid at any of the 685 time-points assayed.’

Perhaps the reviewer did not notice that Fig. 1d and Fig. 1g are plotted with different Y-axis scales? Nevertheless, we find no grounds for commenting on any aberrant behavior of wild-type pYX132 cells based on our data.

Reviewer #3 (Remarks to the Author):

I was asked to specifically look at the proteomic data in the paper.

The data search and phospho site assignment is fine I think (PD, SEQUEST and phosphoRS). These were done by standard methods.

Processing of mass spectrometry (MS) raw files using the SEQUEST algorithm of the Proteome Discoverer (PD) software and allocation of phosphorylation sites using phosphoRS are indeed standard methods. A detailed account of the methods and settings applied was provided in the Supplementary Methods of the original manuscript draft (see Mass Spectrometry, Supplementary Information at the ProteomeXchange Consortium (<http://www.proteomexchange.org>) via the PRoteomics IDentifications (PRIDE) partner repository with the dataset identifier PXD001853. The identifier number was pointed out in the Supplementary Information (page 17, lines 12-14) and at the appropriate place in the Nature Communications Manuscript Tracking System upon submission).

However, there are issues with the reporting of the data and also with replicates. The data tables in the supplement are confusing at best;

On re-inspection of the Supplementary Information we indeed discovered that the data was presented in an inadequate/confusing way. In the revised manuscript we have addressed these issues by restructuring the relevant Figures, Supplementary Figures, Supplementary Tables and corresponding Figure legends (see below). Whereas it is our firm belief that all relevant information for inspection of the MS data was included in the original submission we acknowledge that the way we presented them made it hard to access raw data.

In fact, it was our distinct aim to prepare Supplementary Tables 1 and 2 for the more general readership of Nature Communications and to a lesser extent for MS experts. Furthermore, our intention was to provide a comprehensive overview of the H₂O₂-induced changes in the phosphorylation pattern of the transcription factor Msn2 rather than to focus on one or two specific sites. In order to accomplish this all technical information regarding the MS-analysis was removed on purpose to simplify the reading of the Supplementary Tables.

Nevertheless, we have substantially restructured the MS data presentation in order to make the data easily accessible both to non-specialists and experts. Supplementary Tables 1 and 2 have been fused and are now effectively :

'Supplementary Table 1 - Mass spectrometry results – quantification of Msn2 phosphorylation in response to H₂O₂.'

The table includes 3 worksheets. Worksheet number 1 (overview) lists the average L/H ratios of all Msn2 peptides from all experimental conditions. Worksheet 2 [n=2 (biological replicates)] shows data from key experiments that are covered by two biological replicates. Columns detail results of the individual biological replicates (thereby providing insight in the technical variance) are hidden - for inspection please unhide. Worksheet 3 [n=1 (biological replicates)] shows the data covered by single measurements. The experimental conditions are now color coded and the sample size for each experimental condition is clearly indicated. A detailed description of the presented data is provided for each worksheet.

Supplementary Table 3 was removed and is now effectively :

‘Supplementary Table 2 - Mass spectrometry results – complete peptide spectrum match (PSM) list of Msn2 in response to H₂O₂.’

We have included an experimental (color) code making it easier to filter the data pertaining to a certain experiment (columns A - D). The experiments are labeled in the same color as they are presented in Supplementary Table 1. We have also included further details and explanations on column labels, filter settings etc. on a separate worksheet within Supplementary Table 2. We therefore believe that it is now much easier to follow which PSM contributed to a given SILAC ratio and by extracting the .RAW files name the corresponding raw data can also be inspected should the reader so desire.

On a final note, we have updated Figures 3d-f and Supplementary Figures 3j-l now presenting the data in a more holistic view where SILAC ratios of the biological replicates are shown as individual data points to conform with the journal guidelines. In the updated Figure 3 and Supplementary Figure 3 we also highlight the technical variance in the quantification of phosphorylation in each of these biological replicates. To illustrate the overall high reproducibility of Msn2 phosphorylation site quantifications by our SILAC-based MS analyses we have finally included a scatter plot showing a high correlation between measurements in the two biological replicates (Supplementary Figure 3m).

The organization of the MS supplemental data makes it pretty impossible to do anything but take their word for it (data from several different methods and conditions are combined to provide evidence for each specific site...

We would like to explicitly point out that we did not combine, merge or average SILAC ratios from different experimental conditions, nor did we indicate such a procedure in the manuscript text or in the Supplementary Information.

SILAC ratios of peptide variants (such as different charge states, missed cleavages, etc.) harboring the same phosphorylation site have been integrated within the individual biological replicates. We refer to this procedure in the Supplementary Materials, Supplementary Information, page 17, lines 17-21: "...peptide entries were grouped according to their biological samples (all .raw files of the same sample) and designated as replicates of an experimental condition. Phosphopeptides spanning the same set of phosphorylated residues were further grouped into and designated as a phosphorylation site group (PSG)...".

Thereby obtained values have been averaged between biological replicates (see Supplementary Materials, page 17, lines 23-24: "...PSG and counter-peptide ratios were log₂ transformed and averaged, and a confidence interval of ± 1 SD was calculated.").

*For example, consider the calculations employed to obtain a SILAC-ratio for phospho-Ser²⁸⁸, which has been found in both biological replicates in the experiments with wild-type cells treated with 0.4 mM H₂O₂ for 10 minutes (Supplementary Table 2, applying filters ‘wild-type’, ‘0.4 mM’ and ‘10 min’ to columns A-C (labelled ‘Strain’, ‘[H₂O₂]’ and ‘Time point’) and) ‘S(288)’ as a filter for column I (‘Amino acid position of the phosphorylation site’). By setting an additional filter in column T (spec-file name) to: “rep1_SILAC_Msn2_0,4mM_H2O2_10min_tio2_Reiter_20120507.raw” all peptides covering phosphorylated Ser288 in biological replicate 1 are displayed (in total 19) including 5 peptides for which SILAC ratios were also obtained: namely, RFS#DVITNQFPSM*TNSR (z=2), RFS#DVITNQFPSM*TNSR (z=3), RFS#DVITNQFPSMTNSR (z=2), RRFS#DVITNQFPSM*TNSR (z=3), and RRFS#DVITNQFPSMTNSR (z=3) (#: phosphorylation, *: oxidation). To quantify Msn2 S288 phosphorylation in this biological replicate we determined the geometric mean of these 5 SILAC ratios. The same procedure was applied for replicate 2. Results of replicate 1 and 2 have been averaged (see Supplementary Table 1, sheet ‘n=2 (biological*

replicates)' line 14, column AB and the hidden columns AJ and AR for values in each of the two biological replicates).

Regarding the comment "data from several different methods ... are combined" we are not sure on what grounds the reviewer states this but assume that he/she refers to the fact that MS3 and multi-stage activation (MSA) have been applied for fragmentation. However, we do not consider this a problem as both methods are based on SILAC quantifications of high resolution MS1 spectra and the advantage of using MSA over MS3 mostly lie in an increased proportion of peptides identified (Ulitz P et al, J Proteome Res, 2009). In the PRIDE repository a separate table indicated whether MSA or MS3 was applied for a given measurement.

... and site specific quantitation is separated from phosphosite evidence - meaning we don't know which PSM they are quantifying off of).

We listed the peptide spectrum matches (PSM) that have been selected for quantification of SILAC ratios in Supplementary Table 3 in the original submission and this information can now be found in Supplementary Table 2. In fact, these tables list all identified PSMs (from all MS experiments) and also provided information on phosphorylation site allocation (phosphoRS score/column G), SILAC (L/H) ratio (column J), number of the first scan (column N) and the corresponding raw file name (column T). A selection of further MS-relevant information were additionally provided (columns M-S).

We refer to Supplementary Table 2 in the Supplementary Information section (page 17, lines 11-12): "All identified peptide spectrum matches of Msn2 can be found in Supplementary Table 2.". Furthermore, all technical details and PSMs relevant for our calculation can be accessed in the data deposited on the PRIDE server, as indicated above. Using this information it is also possible to inspect files of the raw data.

They only have one replicate at best two for some conditions and I'm not sure why.

We mostly focused on two PKA-responsive, key regulatory phosphorylation sites of Msn2, namely Ser²⁸⁸ and Ser³⁰⁴, as their phosphorylation status largely determine intracellular localization of Msn2.

In a first series of experiments, we measured the effect of H₂O₂ on the phosphorylation status of these sites in wild-type cells. To do so we chose three different experimental conditions: we exposed wild type cells for 10 minutes to (i) 0.2 mM, (ii) 0.4 mM, and (iii) 0.8 mM H₂O₂, respectively (NB. 0.4 mM H₂O₂ mimics physiological/intracellular levels of H₂O₂ in response to light and was therefore the most relevant condition in this study).

In a second series of experiments we determined effects of 0.4mM H₂O₂ on the PKA-sites of Msn2 in two deletion mutant strain, *bcy1Δ* (iv) and *tsa1Δ* (v).

In addition, we verified that differences in the phosphorylation pattern of Msn2 seen between wild type and *tsa1Δ* cells were only seen following H₂O₂ treatment and not in untreated cells (vi). We regard (i)-(vi) as key experiments of our MS approach.

Furthermore, we performed measurements on single biological replicates (n=1 (biological replicates)) in order to gain further insight into response kinetics by probing earlier and later time points in the response. We are well aware that these data are of limited statistical relevance. Nevertheless, they confirm the trends observed in experimental series (i)-(v), provide valuable controls for the key observations and confirm that we chose the correct time window for measuring.

If one replicate didn't work you can't just remove it for some site data and not remove it for some other site data, as appears to be the case. For "0.4mM H₂O₂ 10min" in Table S2 it is even written replicate 1 and replicate 3... what happened to replicate 2?

We want to point out that we did not "just remove one replicate". Replicate 3 is effectively replicate 2. The labeling is a mistake which originates from a wrong entry for the corresponding sample in the autosampler of the HPLC system. However, since the corresponding .RAW files (available for download at the PRIDE repository) have also been designated replicate 3, we decided to keep the name to facilitate inspection of the data and avoid confusion. However, the experiment is now listed as biological replicate 2 in the revised Supplementary Tables.

Once they know which site(s) they are interested in, they could have re-run these samples with an inclusion list or some other method to make sure they detect it in every replicate.

We wanted to provide a comprehensive overview on the changes in the phosphorylation pattern of Msn2 in response to hydrogen peroxide treatment. Msn2 harbors at least 6 canonical PKA consensus motifs and more PKA-responsive sites with a less strict consensus. Ser²⁸⁸ and Ser³⁰⁴ of the Msn2NES and Ser⁵⁸², Ser⁶²⁰, Ser⁶²⁵, and Ser⁶³³ of the NLS have been attributed with regulating nuclear-cytoplasmic shuttling of Msn2. Even though we mostly focused on Ser²⁸⁸ and Ser³⁰⁴, observations made for the other sites are important; since they are also PKA-dependent sites they confirm observations made for Ser²⁸⁸ and Ser³⁰⁴, especially in replicates where only 1 or none PSM was quantified. To do so we used an established setup outlined in Reiter W et al, Mol Cell Biol, 2013. Regarding the inclusion list we appreciate the suggestion, however, we are confident that our results are conclusive enough for all the sites we are interested in using the present setup.

The standard for "site quantification" would be triplicates and the data as it stands doesn't really reach that standard in some cases. Minimally they would need to say in those cases that the data is from a single experiment.

On inspection of the journal guidelines we did not find a defined standard for "site quantification" in MS experiments. In fact, MS-based approaches to quantify phosphorylation based on a sample number of n=1 or 2 have previously been published in high-impact journals (See 'Examples of MS-based phosphoproteomic analyses in high-impact journals...' below). In particular, multiple recent Nature Communications papers featuring MS-based phosphorylation measurements employ single biological replicates or duplicates (n=1-2) (See 'Examples of MS-based phosphoproteomic analyses in Nature Communications 2014-16...').

At this point it is also appropriate to emphasize that we base our conclusions on several lines of evidence and that key experiments are supported by data from multiple experimental setups. Thus, although the MS data nicely illustrate the PKA-dependence of the regulation of endogenous Msn2 upon the addition of H₂O₂, this important point is backed up also e.g. by immunoblot analyses of immunoprecipitated Msn2.

Nevertheless, all of the above mentioned key MS experiments were covered by two independent biological replicates: (i) wild type - 10 minutes, 0.2 mM H₂O₂, (ii) wild type - 10 minutes, 0.4 mM H₂O₂, (iii) wild type - 10 minutes, 0.8 mM H₂O₂, (iv) bcy1Δ 0.4 mM H₂O₂, (v) tsa1Δ 0.4 mM H₂O₂, and (vi) wild type vs. tsa1Δ (untreated).

*Experiments (i)-(iii) indicated a general decrease in phosphorylation at key PKA-motifs (Ser²⁸⁸, Ser³⁰⁴ and Ser⁶³³) of Msn2 in a dose dependent manner. Several other PKA-dependent phosphorylation sites (Ser⁴⁵¹, Ser⁵⁸², Ser⁶²⁵, etc.; Görner et al., Genes Dev. **12** (1998); Görner et al, EMBO J. **21** (2002); Hao et al., Nature Structural & Molecular*

Biology **19** (2012) ; Reiter et al., *Mol Cell Biol.* **33** (2013)) of Msn2 showed a similar behaviour. These observations are strengthened by immunoblot analysis of immunoprecipitated Msn2 showing a similar decrease in the phosphorylation for Ser²⁸⁸ and Ser³⁰⁴ in response to H₂O₂ (n=3, Fig. 3h). Hence, our overall conclusion that PKA activity is reduced in response to H₂O₂ treatment is supported by four independent experiments each of which features n≥2, biological replicates.

In addition, experiments (iv) and (v) indicate that the phosphorylation pattern of Msn2 is less affected by H₂O₂ treatment in *bcy1Δ* and *tsa1Δ* deletion mutant strains, respectively. Finally, experiments in (vi) indicate no significant difference in Msn2 phosphorylation between unstressed wild type and *tsa1Δ* cells. To validate MS data, we again provided results from quantitative immunoblot analysis on *tsa1Δ* cells (n=3, Fig. 3h).

REFERENCES

Examples of MS-based phosphoproteomic analyses in high-impact journals (n=2)

A Cell-Signaling Network Temporally Resolves Specific versus Promiscuous Phosphorylation

Evgeny Kanshin et al, *Cell Reports*, 2015

<http://www.sciencedirect.com/science/article/pii/S2211124715000777>

Phosphoproteomic analyses reveal novel cross-modulation mechanisms between two signaling pathways in yeast

Stefania Vaga et al, *Molecular Systems Biology*, 2014

<https://www.ncbi.nlm.nih.gov/pmc/articles/PMC4300490/>

quote: *To assess the reproducibility of the measured dynamic profiles and their agreement with published data, NaCl-only and pheromone-only time course experiments were performed in duplicate. The known P-peps indicating activation of Hog1 and Fus3, the two MAPKs of the respective pathways, were detectable with a high degree of reproducibility (Supplementary Fig S1).*

Global Analysis of Cdk1 Substrate Phosphorylation Sites Provides Insights into Evolution
Liam J. Holt et al., *Science* 25, 2009

<http://science.sciencemag.org/content/325/5948/1682.full>

quote: *“ S/T-P motif ...the phosphopeptide must decline in abundance by at least 50% after Cdk1 inhibition (as indicated by $\log_2 H/L < -1$) in one or more of our three experiments. Based on this double filtering, 547 distinct phosphorylation sites were identified on 308 candidate Cdk1 substrates (Fig. 1C and tables S1 and S2, substrate list).”*

Figure 1C Legend venn diagram, quote: *“Venn diagram representing the number of proteins and unique phosphorylation sites identified in the three experiments.”; Table S2*

quote: *“This table lists phosphorylation sites in peptides that declined in abundance by at least 50% following Cdk1 inhibition in one experiment and also declined after Cdk1 inhibition in another experiment”*

hence: all sites of S2 are n=2

Phosphoproteome dynamics of Saccharomyces cerevisiae under heat shock and cold stress
Evgeny Kanshin et al., *Molecular Systems Biology*, 2015

<http://onlinelibrary.wiley.com/doi/10.15252/msb.20156170/full>

quote: *In addition to an FDR of 1% set for peptide, protein and phosphosite identification levels, we used additional criteria to increase data quality. First, we considered only peptides for which abundance ratios (FC = Stimulus/Control) were measured in at least 10 out of 15 time points. Then, we set a cutoff for average phosphosite localization confidence across experiments (time points) to 0.75. Final selection of dynamic profiles was achieved by fitting all temporal curves to a polynomial model and filtering out those fits with a regression coefficient $R^2 < 0.9$ (Supplementary Fig S3 and Supplementary Table S2). Based on these criteria, we obtained 5,554 high-confidence profiles, of which 2,777 represented dynamic phosphosite profiles (Supplementary Table S1). Hence n=1.*

Examples of MS-based phosphoproteomic analyses in Nature Communications 2014-16 (n=1 or 2)

Systematic functional analysis of kinases in the fungal pathogen Cryptococcus neoformans
Kyung-Tae Lee et al., *Nature communications* 7, 2016

<http://www.nature.com/articles/ncomms12766#s7>

2 conditions, n=2, **quote:** Spectral counts were used to estimate relative phosphopeptide abundance between the wild-type and *vrk1Δ* mutant strains. The data were further normalized by dividing each ratio (*vrk1Δ* per wild-type) by the median value of the ratio of each LC-MS/MS run, resulting in normalized relative phosphopeptide abundance. The Student's *t*-test was used to assess the statistically significant difference between the samples.

no MS based quantification:

Aurora-A recruitment and centrosomal maturation are regulated by a Golgi-activated pool of Src during G2

Maria Luisa Barretta et al., *Nature communications* 7, 2016

<http://www.nature.com/articles/ncomms11727>

Identification of Aurora-A-Y148 phosphorylation via MS, quantification via western blot

Quote: 'Representative western blotting of two different experiments...'

Activation of diverse signalling pathways by oncogenic PIK3CA mutations

Xinyan Wu et al, *Nature Communications* 5, 2014

<http://www.nature.com/articles/ncomms5961>

n = 2 (biological replicates)

Quote: '...phosphopeptide ratios (Ex9-KI or Ex20-KI versus MCF10A) from two SILAC reverse-labelled biological replicates...'

Large-scale models of signal propagation in human cells derived from discovery phosphoproteomic data

Camille D. A. Terfve et al, *Nature Communications* 2015

<http://www.nature.com/articles/ncomms9033>

biological duplicates and technical triplicates

Quote: '...Each treatment is performed in biological duplicates (successive early passages),...'

Large-scale determination of absolute phosphorylation stoichiometries in human cells by motif-targeting quantitative proteomics

Chia-Feng Tsai et al, *Nature Communications*, 2015

<http://www.nature.com/articles/ncomms7622>

n=2 biological replicates

Quote: '...By duplicate analysis, 96% averaged efficiency was obtained for the 1,668 identified phosphopeptides in the mixture of phosphatase-treated and -untreated aliquots..'

p38- and MK2-dependent signalling promotes stress-induced centriolar satellite remodelling via 14-3-3-dependent sequestration of CEP131/AZI1

Maxim A. X. Tollenaere et al, *Nature Communications*, 2015

<http://www.nature.com/articles/ncomms10075>

n=1 and quantitative data on MK2 phosphorylation presented

An Arabidopsis PWI and RRM motif-containing protein is critical for pre-mRNA splicing and ABA responses

Xiangqiang Zhan et al, *Nature Communications*, 2015

<http://www.nature.com/articles/ncomms9139>

n=1 and quantitative MS-based phosphorylation data presented

PTEN regulates EG5 to control spindle architecture and chromosome congression during mitosis

Jinxue He et al, *Nature Communications*, 2016

<http://www.nature.com/articles/ncomms12355>

n=1 and quantitative data on EG5 phosphorylation presented

Reviewers' Comments:

Reviewer #3 (Remarks to the Author):

The authors have reorganized the supplemental tables to make it easier to see what they have done and they have renamed "replicate 3" as "Biological Replicate 2", which clarifies this confusion. As such, the authors have addressed my major concerns.